# Harnessing the Universal Geometry of Embeddings

**Rishi Jha**    **Collin Zhang**    **Vitaly Shmatikov**    **John X. Morris**
Department of Computer Science
Cornell University

## Abstract

We introduce the first method for translating text embeddings from one vector space to another without any paired data, encoders, or predefined sets of matches. Our unsupervised approach translates any embedding to and from a universal latent representation (i.e., a universal semantic structure conjectured by the Platonic Representation Hypothesis). Our translations achieve high cosine similarity across model pairs with different architectures, parameter counts, and training datasets.

The ability to translate unknown embeddings into a different space while preserving their geometry has serious implications for security. An adversary with access to a database of only embedding vectors can extract sensitive information about underlying documents, sufficient for classification and attribute inference.

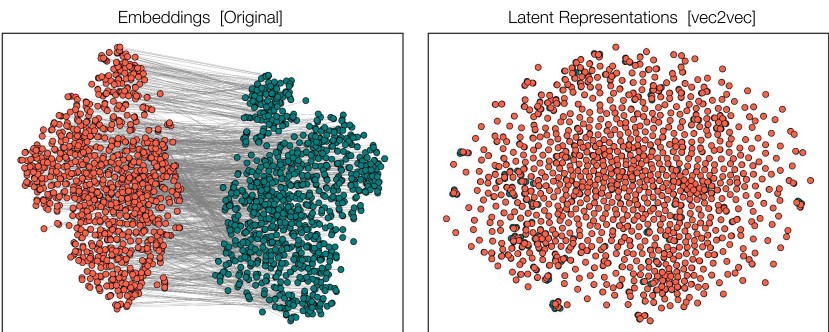

Figure 1: Left: input embeddings from different model families (T5-based GTR [47] and BERT-based GTE [32]) are fundamentally incomparable. Right: given unpaired embedding samples from different models on different texts, our model learns a latent representation where they are closely aligned.

## 1 Introduction

Text embeddings are the backbone of modern NLP, powering tasks like retrieval, RAG, classification, and clustering. There are many embedding models trained on different datasets, data shufflings, and initializations. An embedding of a text encodes its semantics: a good model maps texts with similar semantics to vectors close to each other in the embedding space. Since semantics is a property of text, different embeddings of the same text should encode the same semantics. In practice, however, different models encode texts into completely different and incompatible vector spaces.

The Platonic Representation Hypothesis Huh et al. [18] conjectures that all vision models of sufficient size converge to the same latent representation. We propose a stronger, constructive version of this hypothesis for text models: the universal latent structure of text representations can be learned and, furthermore, harnessed to translate representations from one space to another without any paired data or encoders.

39th Conference on Neural Information Processing Systems (NeurIPS 2025).

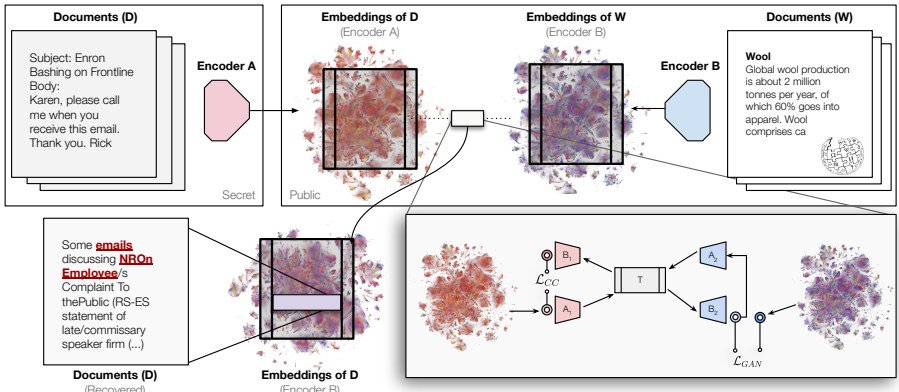

Figure 2: Given only a vector database from an unknown model, vec2vec translates the database into the space of a known model using latent structure alone. Converted embeddings reveal sensitive information about the original documents, such as the topic of an email (pictured, real example).

In this work, we show that the Strong Platonic Representation Hypothesis holds in practice. Given unpaired examples of embeddings from two models with different architectures and training data, our method learns a latent representation in which the embeddings are almost identical (Figure 1).

We draw inspiration from research on aligning word embeddings across languages [62, 10, 15, 9] and unsupervised image translation [36, 70]. Our vec2vec method uses adversarial losses and cycle consistency to learn to encode embeddings into a shared latent space and decode with minimal loss. This makes unsupervised translation possible. We use a basic adversarial approach with vector space preservation [46] to learn a mapping from an unknown embedding distribution to a known one.

vec2vec is the **first method to successfully translate embeddings from the space of one model to another without paired data.**[1] vec2vec translations achieve cosine similarity as high as $0.96$ to the ground-truth vectors in their target embedding spaces and perfect matching on over $8000$ shuffled embeddings (without access to the set of possible matches in advance).

To show that our translations preserve not only the relative geometry of embeddings but also the semantics of underlying inputs, we extract information from them using zero-shot attribute inference and inversion, without any knowledge of the model that produced the original embeddings.[2]

## 2 Problem formulation: unsupervised embedding translation

Consider a collection of embedding vectors $\{u_1, \ldots u_n\}$, for example, a dump of a compromised vector database, where each $u_i = M_1(d_i)$ is generated by an unknown encoder $M_1 : \mathbb{V}^s \to \mathbb{R}^{d_{M_1}}$ from an unknown document $d_i$. We cannot make queries to $M_1$ and do not know its training data, nor architectural details. Our goal is to extract any information about the documents $d_i$.

We do assume access to a different encoder $M_2$ that we can query at will to generate new embeddings in some other space. We also assume high-level distributional knowledge about the hidden documents: their modality (text) and language (e.g., English). To extract information, we may translate $\{u_1, \ldots u_n\}$ into the output space of $M_2$ and apply techniques like inversion that require the encoder.

**Limitations of correspondence methods.** There is significant prior research on the problem of *matching* or *correspondence* between sets of embedding vectors [1, 49, 8, 54]. These methods typically assume that the two (or more) sets of embeddings are generated by different encoders on the *same or highly-overlapping inputs*. In other words, for each unknown vector, there must already exist a set of candidate vectors in a different embedding. In practice, it is unrealistic to expect that such a database be available, so these methods are not directly applicable. Some matching methods,

---

[1]Prior work has successfully translated *word* embeddings between languages, typically relying on overlapping vocabularies across languages. In contrast, we translate embeddings of entire sequences between model spaces.

[2]Our code is available on GitHub.

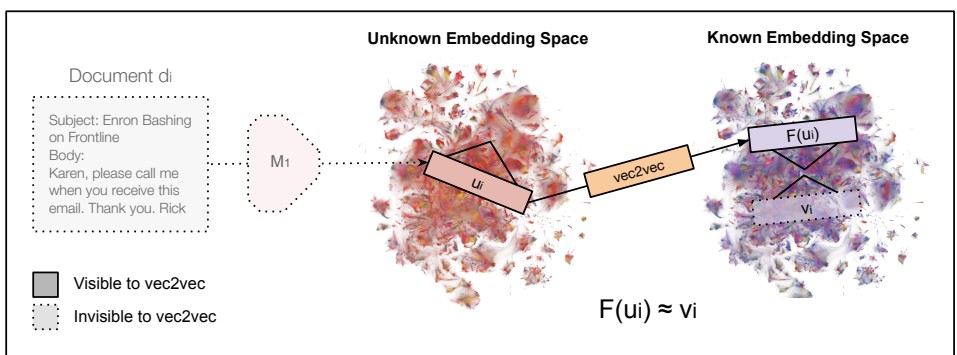

Figure 3: **Unsupervised embedding translation**. With access to only $u_i = M_1(d_i)$, vec2vec seeks to generate a translation $F(u_i)$ that is close in $M_2$'s embedding space to the ideal embedding $v_i = M_2(d_i)$ without access to $d_i$, $v_i$, or $M_1$.

however, support translation between embedding spaces without overlapping inputs. Our experiments demonstrate that these methods struggle significantly, even when correspondence exists.

Our task is inherently more challenging than matching, because we do not assume access to encoder $M_1$, nor do we have additional representations of documents $d_1, \ldots, d_n$ beyond their embeddings $u_i = M_1(d_i)$. Therefore, we rely solely on unsupervised *translation* from $M_1$ to $M_2$. The effectiveness of such unsupervised translation approaches thus critically depends on identifying and leveraging shared geometric structures within the embedding spaces produced by $M_1$ and $M_2$.

**The Strong Platonic Representation Hypothesis.** Our hope that unsupervised embedding translation is possible at all rests on the stronger version of the Platonic Representation Hypothesis [18]. Our conjecture is as follows: *neural networks trained with the same objective and modality, but with different data and model architectures, converge to a universal latent space such that a translation between their respective representations can be learned without any pairwise correspondence.*

**Translation enables information extraction.** Solving unsupervised translation will allow us to use information extraction tools designed to operate on vectors produced by known encoders. For example, we could apply inversion models [43, 67] to recover unknown documents $\{d_i\}$.

## 3 Our method: vec2vec

Unsupervised translation has been successful in computer vision, using a combination of cycle consistency and adversarial regularization [36, 70]. Our design of vec2vec is inspired in part by these methods. We aim to learn embedding-space translations that are cycle-consistent (mapping to and from an embedding space should end in the same place) and indistinguishable (embeddings for the same text from either space should have identical latents).

### 3.1 Architecture

We propose a modular architecture, where embeddings are encoded and decoded using space-specific adapter modules and passed through a shared backbone network. Figure 2 shows these components. Input adapters $A_1 : \mathbb{R}^d \to \mathbb{R}^Z$ and $A_2 : \mathbb{R}^d \to \mathbb{R}^Z$ transform embeddings from each encoder-specific space into a universal latent representation of dimension $Z$. The shared backbone $T : \mathbb{R}^Z \to \mathbb{R}^Z$ extracts a common latent embedding from adapted inputs. Output adapters $B_1 : \mathbb{R}^Z \to \mathbb{R}^d$ and $B_2 : \mathbb{R}^Z \to \mathbb{R}^d$ translate these common latent embeddings back into the encoder-specific spaces. Thus, translation functions $F_1, F_2$ and additional reconstruction mappings $R_1, R_2$ are defined as:

$$F_1 = B_2 \circ T \circ A_1, \quad F_2 = B_1 \circ T \circ A_2 \quad R_1 = B_1 \circ T \circ A_1 \quad R_2 = B_2 \circ T \circ A_2$$

Parameters of all components are collectively denoted $\theta = \{A_1, A_2, T, B_1, B_2\}$.

Unlike images, embeddings do not have any spatial bias. Instead of CNNs, we use multilayer perceptrons (MLP) with residual connections, layer normalization, and SiLU nonlinearities. Discriminators mirror this structure but omit residual connections to simplify adversarial learning.

## 3.2 Optimization

In addition to the 'generator' networks $F$ and $R$, we introduce discriminators operating on both the latent representations of $F$ ($D_1^\ell, D_2^\ell$) and the output embeddings ($D_1, D_2$).

Our goal is to train the parameters of $\theta$ by solving:

$$\theta^* = \arg\min_\theta \max_{D_1,D_2,D_1^\ell,D_2^\ell} \mathcal{L}_{\text{adv}}(F_1, F_2, D_1, D_2, D_1^\ell, D_2^\ell) + \lambda_{\text{gen}}\mathcal{L}_{\text{gen}}(\theta), \tag{1}$$

where $\mathcal{L}_{\text{adv}}$ and $\mathcal{L}_{\text{gen}}$ represent adversarial and generator-specific constraints respectively and hyperparameter $\lambda_{\text{gen}}$ controls their tradeoff.

**Adversarial.** The adversarial loss encourages generated embeddings to match the empirical distributions of original embeddings both at the embedding and latent levels. Specifically, applying the standard GAN loss formulation [13] to both levels yields:

$$\mathcal{L}_{\text{adv}}(F_1, F_2, D_1, D_2, D_1^\ell, D_2^\ell) = \mathcal{L}_{\text{GAN}}(D_1, F_1) + \mathcal{L}_{\text{GAN}}(D_2, F_2)$$
$$+ \mathcal{L}_{\text{GAN}}(D_1^\ell, T \circ A_1) + \mathcal{L}_{\text{GAN}}(D_2^\ell, T \circ A_2).$$

**Generator.** Because adversarial losses alone do not guarantee that translated embeddings preserve semantics [70], we introduce three additional constraints to help the generator learn a useful mapping:

*Reconstruction* enforces that an embedding, when mapped into the latent space and back into its original embedding space, closely matches its initial representation:

$$\mathcal{L}_{\text{rec}}(R_1, R_2) = \mathbb{E}_{x \sim p}\|R_1(x) - x\|_2^2 + \mathbb{E}_{y \sim q}\|R_2(y) - y\|_2^2.$$

where $p$ and $q$ are distributions of embeddings sampled from $M_1$ and $M_2$, respectively.

*Cycle-consistency* acts as an unsupervised proxy for supervised pair alignment, ensuring that $F$ and $G$ can translate an embedding to the other embedding space and back again with minimal corruption:

$$\mathcal{L}_{\text{CC}}(F_1, F_2) = \mathbb{E}_{x \sim p}\|F_2(F_1(x)) - x\|_2^2 + \mathbb{E}_{y \sim q}\|F_1(F_2(y)) - y\|_2^2.$$

*Vector space preservation (VSP)* ensures that pairwise relationships between translated embeddings are consistent with the target space [46, 65]. Given a batch of $B$ embeddings $x_1, ..., x_B$ and $y_1, ..., y_B$, we sum their average pairwise distances after translation by both $F_1$ and $F_2$:

$$\mathcal{L}_{\text{VSP}}(F_1, F_2) = \frac{1}{B^2}\sum_{i=1}^{B}\sum_{j=1}^{B}\left[\|M_1(x_i) \cdot M_1(x_j) - F_2(M_2(y_i)) \cdot F_2(M_2(y_j))\|_2^2\right.$$
$$\left. + \|M_2(y_i) \cdot M_2(y_j) - F_1(M_1(x_i)) \cdot F_1(M_1(x_j))\|_2^2\right]$$

Combining these losses yields: $\mathcal{L}_{\text{gen}}(\theta) = \lambda_{\text{rec}}\mathcal{L}_{\text{rec}}(R_1, R_2) + \lambda_{\text{CC}}\mathcal{L}_{\text{CC}}(F_1, F_2) + \lambda_{\text{VSP}}\mathcal{L}_{\text{VSP}}(F_1, F_2)$, where hyperparameters $\lambda_{\text{CC}}$, $\lambda_{\text{rec}}$, and $\lambda_{\text{VSP}}$ control relative importance.

# 4 Experimental setup

## 4.1 Preliminaries

**Datasets.** We use the *Natural Questions (NQ)* [25] dataset of user queries and Wikipedia-sourced answers for training (a 2-million subset) and evaluation (a 65536 subset). To evaluate information extraction, we use *TweetTopic* [2], a dataset of tweets multi-labeled by 19 topics; a random 8192-record subset of *Pseudo Re-identified MIMIC-III (MIMIC)* [28], a pseudo re-identified version of the MIMIC dataset [19] of patient records multi-labeled by 2673 MedCAT [24] disease descriptions; and a random 50-email subset of the *Enron Email Corpus (Enron)* [21], an unlabeled, public dataset of internal emails from a defunct energy company. In Appendix D, we ablate a model on *MS COCO* [34], a captioned image dataset, to evaluate performance on multimodal retrieval.

**Models.** Table 1 lists the embedding models representing four size categories, five transformer backbones, and two output dimensionalities. Granite is multilingual; CLIP is multimodal. Since Qwen is very compute-intensive, we only evaluate it for a single model pair in Appendix C.

| Model | Params (M) | Backbone | Year | Dims | Max Seq. |
|---|---|---|---|---|---|
| [47] gtr | 110 | T5 | 2021 | 768 | 512 |
| [50] clip | 151 | CLIP | 2021 | 512 | 77 |
| [58] e5 | 109 | BERT | 2022 | 768 | 512 |
| [32] gte | 109 | BERT | 2023 | 768 | 512 |
| [68] stella | 109 | BERT | 2023 | 768 | 512 |
| [14] granite | 278 | RoBERTa | 2024 | 768 | 512 |
| [69] qwen | 4000 | Qwen3 | 2025 | 2560 | 32K |

Table 1: Embedding models used in our experiments.

**Training.** Unless otherwise specified, each `vec2vec` is trained on two sets of embeddings generated from disjoint sets of 1 million 64-token sequences sampled from NQ (see Section 7 for experiments with fewer embeddings). Due to GAN instability [53], we select the best of multiple initializations (see Appendix E) and leave more robust training to future work. See Appendix A for compute details.

## 4.2 Evaluating translation

Let $u_i = M_1(d_i)$ and $v_i = M_2(d_i)$ denote the source and target embeddings of the same input $d_i$. The goal of translation is to generate a vector that is as close to $v_i$ as possible. We say that $(u_i, v_j)$ are "aligned" by the translator $F$ if $v_j$ is the closest embedding to $F(u_i)$: $j = \arg\min_k \cos\big(F(u_i), v_k\big)$. A perfect translator $F^*$ satisfies $i = \arg\min_k \cos\big(F^*(u_i), v_k\big)$ for all $i$.

Given (unknown) embeddings $\{M_2(d_j)\}_{j=0}^n$ ordered by decreasing cosine similarity to $F(u_i)$, let $r_i$ be the rank of the correct embedding $v_i = M_2(d_i)$. To measure quality of $F$, we use three metrics. **Mean Cosine Similarity** measures how close translations are, on average, to their targets. **Top-1 Accuracy** is the fraction of translations whose target is closer than any other embedding. **Mean Rank** is the average rank of targets with respect to translations. The ideal translator $F^*$ achieves mean similarity of $1.0$, top-1 accuracy of $1.0$, and mean rank of $1.0$. Recall that a random alignment corresponds to a mean rank of $\frac{n}{2}$. Formally,

$$\cos(u_i, v_i) = \frac{1}{n}\sum_{i=1}^n \big[1 - \cos\big(F(u_i), v_i\big)\big] \quad \text{Top-1}(r) = \frac{1}{n}\sum_{i=1}^n \mathbf{1}\{r_i = 1\} \quad \text{Rank(r)} = \frac{1}{n}\sum_{i=1}^n r_i$$

`vec2vec` is the first unsupervised embedding translator, thus there is no direct baseline. As our **Naïve** baseline, we simply use $F(x) = x$ to measure geometric similarity between embedding spaces. The second (pseudo)baseline is **Oracle-aided optimal transport**. It assumes that candidate targets are known and is thus strictly easier than `vec2vec` and the Naïve baseline. We solve optimal assignment, $\pi^* = \arg\min_\pi \sum_{i=1}^n \cos(u_i, v_{\pi(i)})$, via either the Hungarian, Earth Mover's Distance, Sinkhorn, or (Entropic) Gromov-Wasserstein algorithms, choosing the solver with the lowest rank for each experiment. See Appendix B for more details.

## 4.3 Evaluating information extraction

We measure whether translation preserves semantics via *attribute inference*: for each translated embedding $F(M_1(d_i))$, our goal is to infer attributes $c_i \subseteq \mathcal{C}$ of $d_i$.

The first method we use is **zero-shot embedding attribute inference**: calculate pairwise cosine similarities between $F(M_1(d_i))$ and the embeddings of all attributes in $\mathcal{C}$, identify top $k$ closest attributes, and measure whether they are correct via *top-k accuracy*: $\frac{1}{n}\sum_{i=0}^n \mathbf{1}\big\{|c_i^k \cap c_i| \geq 1\big\}$.

The second method is **embedding inversion** that recovers text inputs from embeddings. Since [43] requires a pre-trained inversion model for each embedding space, we use [67] instead to generate an approximation $d_i'$ of $d_i$ from $F(M_1(d_i))$ in a zero-shot manner. We measure the extracted information using *LLM judge accuracy*: the fraction of translated embeddings for which GPT-4o determines that $d'$ reveals information in $d$. See Appendix H for our prompt.

In addition to the Naïve baseline, we also consider an **Oracle attribute inference**: zero-shot classification with the correct embedding $M_2(d)$ and class labels $M_2(\mathcal{C})$.

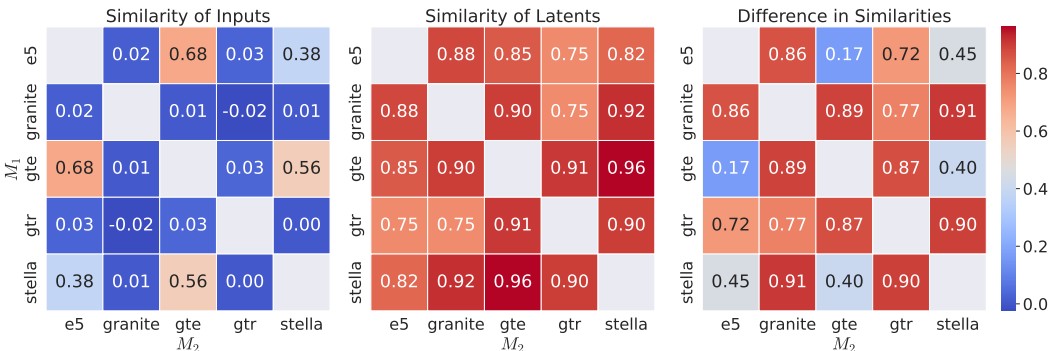

Figure 4: Pairwise cosine similarities of input embeddings (left) and their `vec2vec` latents (middle) across different embedding pairs. The absolute difference between the heatmaps plots is on the right. All numbers are computed on the same batch of 1024 NQ texts.

## 5 `vec2vec` learns to translate embeddings without any paired data

We first show that `vec2vec` learns a universal latent space, then demonstrate that this space preserves the geometry of all embeddings. Therefore, we can use it like a **universal language of text encoders** to translate their representations without any paired data.

`vec2vec` **learns a universal latent space.** `vec2vec` projects embeddings $M_{1,2,...}$ into a shared latent space via compositions of input adapters ($A_{1,2,...}$) and a shared translator $T$. Figure 4 shows that even when the embeddings $u_i = M_1(d_i)$ and $v_i = M_2(d_i)$ are far apart (*i.e.,* have low cosine similarity), their representations in `vec2vec`'s latent space are incredibly close: $T(A_1(u_i)) \approx T(A_2(v_i))$. Figure 1 visualizes this (via two-dimensional projections) for `vec2vec` trained on GTE and GTR embeddings: the embeddings are far apart, but their latents are *nearly overlapping*.

| | | vec2vec | | | Naïve Baseline | | | OT Baseline | | |
|---|---|---|---|---|---|---|---|---|---|---|
| $M_1$ | $M_2$ | cos($\cdot$) ↑ | T-1 ↑ | Rank ↓ | cos($\cdot$) ↑ | T-1 ↑ | Rank ↓ | cos($\cdot$) ↑ | T-1 ↑ | Rank ↓ |
| gra. | gtr | **0.80** (0.0) | **0.99** | **1.19** (0.1) | -0.03 (0.0) | 0.00 | 4168.73 (9.2) | 0.70 (0.0) | 0.00 | 2773.72 (8.6)[‡] |
| | gte | **0.87** (0.0) | **0.95** | **1.18** (0.0) | 0.01 (0.0) | 0.00 | 4088.58 (9.2) | 0.85 (0.0) | 0.00 | 2680.02 (8.6)[‡] |
| | ste. | **0.79** (0.0) | **0.98** | **1.05** (0.0) | 0.01 (0.0) | 0.00 | 4208.26 (9.2) | 0.67 (0.0) | 0.00 | 3446.52 (8.8)[‡] |
| | e5 | **0.85** (0.0) | **0.98** | **1.11** (0.0) | 0.02 (0.0) | 0.00 | 4111.60 (9.2) | 0.83 (0.0) | 0.00 | 3569.59 (8.7)[‡] |
| gtr | gra. | **0.81** (0.0) | **0.99** | **1.02** (0.0) | -0.03 (0.0) | 0.00 | 4169.76 (9.2) | 0.70 (0.0) | 0.00 | 2775.17 (8.6)[‡] |
| | gte | **0.87** (0.0) | **0.93** | **2.31** (0.1) | 0.04 (0.0) | 0.00 | 4080.92 (9.2) | 0.85 (0.0) | 0.00 | 3070.69 (8.9)[‡] |
| | ste. | **0.80** (0.0) | **0.99** | **1.03** (0.0) | 0.00 (0.0) | 0.00 | 4198.78 (9.2) | 0.67 (0.0) | 0.00 | 3559.06 (9.1)[‡] |
| | e5 | **0.83** (0.0) | **0.84** | **2.88** (0.2) | 0.03 (0.0) | 0.00 | 4082.84 (9.2) | **0.83** (0.0) | 0.00 | 3888.01 (8.9)[‡] |
| gte | gra. | **0.75** (0.0) | **0.95** | **1.22** (0.0) | 0.01 (0.0) | 0.00 | 4079.81 (9.3) | 0.69 (0.0) | 0.00 | 2664.38 (8.6)[‡] |
| | gtr | **0.75** (0.0) | **0.91** | **2.64** (0.1) | 0.04 (0.0) | 0.00 | 4084.15 (9.2) | 0.70 (0.0) | 0.00 | 3064.16 (8.9)[‡] |
| | ste. | **0.89** (0.0) | **1.00** | **1.00** (0.0) | 0.56 (0.0) | **1.00** | **1.00** (0.0) | 0.71 (0.0) | **1.00** | **1.00** (0.0)[†] |
| | e5 | **0.87** (0.0) | 0.99 | 5.19 (0.5) | 0.68 (0.0) | **1.00** | **1.00** (0.0) | 0.84 (0.0) | **1.00** | **1.00** (0.0)[†] |
| ste. | gra. | **0.80** (0.0) | **0.98** | **1.08** (0.0) | 0.01 (0.0) | 0.00 | 4209.08 (9.3) | 0.69 (0.0) | 0.00 | 3419.44 (8.8)[‡] |
| | gtr | **0.82** (0.0) | **1.00** | **1.10** (0.0) | 0.00 (0.0) | 0.00 | 4192.31 (9.2) | 0.70 (0.0) | 0.00 | 3555.64 (9.0)[‡] |
| | gte | **0.92** (0.0) | **1.00** | **1.00** (0.0) | 0.56 (0.0) | **1.00** | **1.00** (0.0) | 0.87 (0.0) | **1.00** | **1.00** (0.0)[†] |
| | e5 | **0.86** (0.0) | **1.00** | **1.00** (0.0) | 0.38 (0.0) | 0.99 | 1.03 (0.0) | 0.83 (0.0) | **1.00** | **1.00** (0.0)[†] |
| e5 | gra. | **0.81** (0.0) | **0.99** | 2.20 (0.2) | 0.02 (0.0) | 0.00 | 4120.60 (9.3) | 0.69 (0.0) | 0.00 | 3526.02 (8.7)[‡] |
| | gtr | **0.74** (0.0) | **0.82** | 2.56 (0.0) | 0.03 (0.0) | 0.00 | 4080.76 (9.3) | 0.70 (0.0) | 0.00 | 3877.03 (8.8)[‡] |
| | gte | **0.90** (0.0) | **1.00** | 1.01 (0.0) | 0.68 (0.0) | **1.00** | **1.00** (0.0) | 0.86 (0.0) | **1.00** | **1.00** (0.0)[†] |
| | ste. | **0.78** (0.0) | **1.00** | **1.00** (0.0) | 0.38 (0.0) | **1.00** | **1.00** (0.0) | 0.69 (0.0) | **1.00** | **1.00** (0.0)[†] |

Table 2: In-distribution translations: `vec2vecs` trained on NQ and evaluated on a 65536 text subset of NQ (chunked in batches of size 8192). The rank metric varies from 1 to 8192, thus 4096 corresponds to a random ordering. Standard errors are shown in parentheses. Bold denotes best value. Symbols denote the lowest-rank solver for specific experiments: Sinkhorn[†] and Gromov-Wasserstein[‡].

| $M_1$ | $M_2$ | TweetTopic | | | MIMIC | | |
|---|---|---|---|---|---|---|---|
| | | cos($\cdot$) ↑ | T-1 ↑ | Rank ↓ | cos($\cdot$) ↑ | T-1 ↑ | Rank ↓ |
| gran. | gtr | 0.74 (0.0) | 0.99 | 1.09 (0.1) | 0.74 (0.0) | 0.60 | 23.38 (1.6) |
| | gte | 0.85 (0.0) | 0.95 | 1.26 (0.1) | 0.85 (0.0) | 0.08 | 346.21 (7.8) |
| | stel. | 0.77 (0.0) | 0.96 | 1.11 (0.0) | 0.72 (0.0) | 0.13 | 242.23 (6.1) |
| | e5 | 0.83 (0.0) | 0.87 | 3.10 (0.7) | 0.84 (0.0) | 0.12 | 361.06 (8.7) |
| gtr | gran. | 0.79 (0.0) | 0.98 | 2.41 (0.6) | 0.78 (0.0) | 0.51 | 35.27 (1.9) |
| | gte | 0.85 (0.0) | 0.96 | 1.29 (0.2) | 0.84 (0.0) | 0.12 | 279.56 (6.9) |
| | stel. | 0.77 (0.0) | 0.96 | 1.10 (0.0) | 0.72 (0.0) | 0.27 | 127.92 (4.4) |
| | e5 | 0.80 (0.0) | 0.53 | 13.38 (1.2) | 0.82 (0.0) | 0.01 | 1413.80 (18.3) |
| gte | gran. | 0.73 (0.0) | 0.94 | 1.33 (0.1) | 0.73 (0.0) | 0.09 | 342.15 (7.8) |
| | gtr | 0.71 (0.0) | 0.95 | 1.29 (0.1) | 0.69 (0.0) | 0.12 | 256.63 (6.4) |
| | stel. | 0.86 (0.0) | 1.00 | 1.00 (0.0) | 0.85 (0.0) | 1.00 | 1.00 (0.0) |
| | e5 | 0.83 (0.0) | 0.91 | 1.57 (0.2) | 0.86 (0.0) | 0.54 | 17.71 (0.9) |
| stel. | gran. | 0.79 (0.0) | 0.99 | 1.09 (0.1) | 0.77 (0.0) | 0.14 | 221.95 (5.9) |
| | gtr | 0.77 (0.0) | 1.00 | 1.00 (0.0) | 0.75 (0.0) | 0.56 | 17.70 (1.0) |
| | gte | 0.90 (0.0) | 1.00 | 1.00 (0.0) | 0.91 (0.0) | 1.00 | 1.00 (0.0) |
| | e5 | 0.85 (0.0) | 0.98 | 1.05 (0.0) | 0.85 (0.0) | 0.51 | 26.33 (1.2) |
| e5 | gran. | 0.79 (0.0) | 0.98 | 1.08 (0.0) | 0.78 (0.0) | 0.21 | 151.09 (4.6) |
| | gtr | 0.67 (0.0) | 0.80 | 3.10 (0.6) | 0.66 (0.0) | 0.01 | 1029.64 (14.9) |
| | gte | 0.87 (0.0) | 0.99 | 1.02 (0.0) | 0.87 (0.0) | 0.60 | 32.59 (2.6) |
| | stel. | 0.75 (0.0) | 0.98 | 1.06 (0.0) | 0.75 (0.0) | 0.46 | 32.12 (1.4) |

Table 3: Out-of-distribution translations: `vec2vecs` trained on NQ and evaluated on the entire TweetTopic test set (800 tweets) and an 8192-record subset of MIMIC. The rank metric varies from 1 to 800 (for TweetTopic) and 8192 (for MIMIC), thus 400 and, respectively, 4096 correspond to a random ordering. Standard errors are shown in parentheses.

`vec2vec` **translations mirror target geometry.** Table 2 shows that `vec2vec` generates embeddings with near-optimal assignment across model pairs, achieving cosine similarity scores up to 0.92, top-1 accuracies up to 100%, and ranks as low as 1. In same-backbone pairings (e.g., (gte, e5)), `vec2vec`'s top-1 accuracy and rank are comparable to both the naïve baseline and (surprisingly) the oracle-aided optimal transport. Although the embeddings generated by `vec2vec` are significantly closer to the ground truth than the naïve baseline, in same-backbone pairings the embeddings are close enough to be compatible. In cross-backbone pairings, `vec2vec` is far superior on all metrics, while baseline methods perform similarly to random guessing.

Table 3 shows that this performance extends to out-of-distribution data. Our `vec2vec` translators were trained on NQ (drawn from Wikipedia), yet exhibit high cosine similarity, high accuracy, and low rank when evaluated on tweets (which are far more colloquial and use emojis) and medical records (which contain domain-specific jargon unlikely to appear in NQ). In Appendix F, we show that baseline methods fail on cross-backbone embedding pairs.

| $M_1$ | $M_2$ | vec2vec | | | OT Baseline | | |
|---|---|---|---|---|---|---|---|
| | | cos($\cdot$) ↑ | T-1 ↑ | Rank ↓ | cos($\cdot$) ↑ | T-1 ↑ | Rank ↓ |
| gra. | clip | **0.78 (0.0)** | **0.35** | **226.62 (3.2)** | 0.76 (0.0) | 0.00 | 4073.58 (9.4)[‡] |
| gtr | | **0.73 (0.0)** | **0.13** | **711.23 (5.9)** | 0.59 (0.0) | 0.00 | 4096.78 (9.2)[‡] |
| gte | | 0.62 (0.0) | **0.00** | **3233.41 (9.8)** | **0.76 (0.0)** | 0.00 | 4026.96 (9.4)[‡] |
| ste. | | **0.77 (0.0)** | **0.31** | **286.69 (3.6)** | 0.76 (0.0) | 0.00 | 3955.71 (8.9)[‡] |
| e5 | | 0.64 (0.0) | **0.01** | **2568.21 (9.4)** | **0.77 (0.0)** | 0.00 | 3771.52 (9.1)[‡] |
| clip | gra. | **0.74 (0.0)** | **0.72** | **4.46 (0.1)** | 0.69 (0.0) | 0.00 | 4053.11 (9.4)[‡] |
| | gtr | **0.67 (0.0)** | **0.27** | **155.11 (2.1)** | 0.49 (0.0) | 0.00 | 4096.35 (9.2)[‡] |
| | gte | 0.75 (0.0) | **0.00** | **2678.90 (8.9)** | **0.85 (0.0)** | 0.00 | 4025.81 (9.3)[‡] |
| | ste. | **0.72 (0.0)** | **0.61** | **22.50 (0.5)** | 0.67 (0.0) | 0.00 | 3951.73 (8.9)[‡] |
| | e5 | 0.73 (0.0) | **0.01** | **1692.28 (8.2)** | **0.83 (0.0)** | 0.00 | 3771.38 (9.0)[‡] |

Table 4: Translations between unimodal and multimodal (CLIP) embeddings: `vec2vecs` trained on NQ and evaluated on a 65536 text subset of NQ (chunked in batches of size 8192). Rank varies from 1 to 8192, thus 4096 corresponds to a random ordering. Since the embedding dimensionalities are different, only the Gromov-Wasserstein[‡] OT baseline is run and the naive baseline does not apply. Bold denotes best value.

| $M_1$ | $M_2$ | TweetTopic ($k=1$) | | | | MIMIC ($k=10$) | | | |
|---|---|---|---|---|---|---|---|---|---|
| | | vec2vec | Naïve | $M_1$ | $M_2$ | vec2vec | Naïve | $M_1$ | $M_2$ |
| gran. | gtr | 0.25 | 0.10 | 0.30 | 0.24 | 0.19 | 0.11 | 0.76 | 0.88 |
| | gte | 0.32 | 0.09 | 0.30 | 0.34 | 0.36 | 0.13 | 0.76 | 1.00 |
| | stel. | 0.24 | 0.10 | 0.30 | 0.28 | 0.27 | 0.04 | 0.76 | 0.96 |
| | e5 | 0.31 | 0.18 | 0.30 | 0.31 | 0.19 | 0.20 | 0.76 | 0.97 |
| gtr | gran. | 0.34 | 0.08 | 0.24 | 0.30 | 0.16 | 0.12 | 0.88 | 0.76 |
| | gte | 0.33 | 0.13 | 0.24 | 0.34 | 0.28 | 0.05 | 0.88 | 1.00 |
| | stel. | 0.30 | 0.10 | 0.24 | 0.28 | 0.25 | 0.07 | 0.88 | 0.96 |
| | e5 | 0.30 | 0.04 | 0.24 | 0.31 | 0.09 | 0.09 | 0.88 | 0.97 |
| gte | gran. | 0.37 | 0.04 | 0.34 | 0.30 | 0.18 | 0.11 | 1.00 | 0.76 |
| | gtr | 0.24 | 0.13 | 0.34 | 0.24 | 0.10 | 0.03 | 1.00 | 0.88 |
| | stel. | 0.31 | 0.20 | 0.34 | 0.28 | 0.68 | 0.83 | 1.00 | 0.96 |
| | e5 | 0.37 | 0.30 | 0.34 | 0.31 | 0.37 | 0.63 | 1.00 | 0.97 |
| stel. | gran. | 0.35 | 0.07 | 0.28 | 0.30 | 0.23 | 0.09 | 0.96 | 0.76 |
| | gtr | 0.26 | 0.13 | 0.28 | 0.24 | 0.22 | 0.09 | 0.96 | 0.88 |
| | gte | 0.38 | 0.36 | 0.28 | 0.34 | 0.90 | 0.98 | 0.96 | 1.00 |
| | e5 | 0.35 | 0.34 | 0.28 | 0.31 | 0.38 | 0.46 | 0.96 | 0.97 |
| e5 | gran. | 0.33 | 0.15 | 0.31 | 0.30 | 0.14 | 0.07 | 0.97 | 0.76 |
| | gtr | 0.26 | 0.22 | 0.31 | 0.24 | 0.11 | 0.04 | 0.97 | 0.88 |
| | gte | 0.34 | 0.28 | 0.31 | 0.34 | 0.47 | 0.66 | 0.97 | 1.00 |
| | stel. | 0.26 | 0.16 | 0.31 | 0.28 | 0.36 | 0.40 | 0.97 | 0.96 |

Table 5: Information leakage via top-$k$ zero-shot attribute inference: vec2vecs trained on NQ and evaluated on the TweetTopic test set (800 tweets) and an 8192-record subset of MIMIC. $M_1$ and $M_2$ represent *ideal* zero-shot inference: attributes and embeddings are encoded using the same model.

Finally, Table 4 shows that vec2vec can even translate to and from the space of CLIP, a multimodal embedding model which was trained in part on *image* data. While the translations are not as strong as in Table 2, vec2vec consistently outperforms the optimal transport baseline. These results show the promise of our method at adapting to new modalities: in particular, the embedding space of CLIP has been successfully connected to other modalities such as heatmaps, audio, and depth charts [12].

## 6   Using vec2vec **translations to extract information**

In this section, we show that vec2vec translations not only preserve the geometric structure of embeddings but also retain sufficient semantics to enable attribute inference.

**Zero-shot attribute inference.**    Table 5 shows that attribute inference on vec2vec translations consistently outperforms the naïve baseline and often does better than the ideal zero-shot baseline which performs inference on ground-truth document and attribute embeddings in the same space (this baseline is imaginary since these embeddings are not available in our setting).

vec2vec translations even work for embeddings of medical records, which are much further from the training distribution than tweets. The attributes in this case are MedCAT disease descriptions, very few of which occur in the training data. Attribute inference on translated embeddings is comparable to the naïve baseline in same-backbone pairings and outperforms it (often greatly) in cross-backbone pairings. The fact that vec2vec preserves the semantics of concepts like "alveolar periostitis" (which never appears in its training data) is evidence that its latent space is indeed a universal representation.

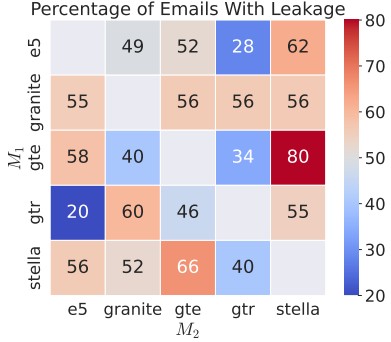

Figure 5: Leakage of information via inversion. Trained on NQ and evaluated on a 50-email subset of the Enron Email Corpus. Cells denote judge accuracy.

**Zero-shot inversion.** Inversion, i.e., reconstruction of text inputs, is more ambitious than attribute inference. vec2vec translations retain enough semantic information that off-the-shelf, zero-shot inversion methods like [67], developed for embeddings computed by standard encoders, extract

> **Ground Truth:** "Subject: `Enron` `Bashing` on Frontline \n Body:..."
>
> **Generation:** "Some emails discussing `NROn` Employee/s `Complaint To thePublic` ..."
>
> **Ground Truth:** "Subject: `Trades for 3/1/02` \n Body: \n `John` , \n The following trades..."
>
> **Generation:** "... `future transactions` may await `John` G..."
>
> **Ground Truth:** " `The following expense report` is ready for approval..."
>
> **Generation:** " `The upcoming expense statement` from YYYY MM Dec..."

Figure 6: Examples of Enron Email Corpus inversions that infer `entities` and `content` .

information for as many as 80% of emails and 67% of tweets given *only* their translated embeddings, for some model pairs (Figure 5 and Appendix G). These inversions are imperfect and we leave development of specialized inverters for translated embeddings to future work. Nevertheless, as exemplified in Figure 6, they still extract potentially sensitive information such as individual and company names, dates, promotions, financial information, outages, and even lunch orders. In Appendix H, we show the prompt we use to measure extraction.

## 7  Ablations

| Method | $\cos(\cdot)\uparrow$ | T-1 $\uparrow$ | Rank $\downarrow$ |
|---|---|---|---|
| `vec2vec` | 0.75 (0.0) | 0.91 | 2.64 (0.1) |
| Naïve Baseline | 0.04 (0.0) | 0.00 | 4084.15 (9.2) |
| OT Baseline | 0.70 (0.0) | 0.00 | 3064.16 (8.9) |
| – VSP loss | 0.58 (0.0) | 0.00 | 4196.64 (9.2) |
| – CC loss | 0.50 (0.0) | 0.00 | 3941.36 (9.3) |
| – latent GAN | 0.49 (0.0) | 0.00 | 3897.09 (9.5) |
| – VSP *and* CC loss | 0.47 (0.0) | 0.00 | 3365.24 (9.3) |
| – hyperparam. tuning | 0.50 (0.0) | 0.00 | 4011.73 (9.3) |

Table 6: gte $\rightarrow$ gtr translators trained without individual components of our method on NQ and evaluated on a 65536-text subset of NQ (chunked in batches of 8192). The rank metric varies from 1 to 8192, thus 4096 corresponds to a random ordering. Standard errors are shown in parentheses.

**Each component of our method is important.** We ablate our method subtractively, measuring the key metrics after removing individual components of our algorithm (described in Section 3). Table 6 shows that each component appears to be *critical* to building good translations. While `vec2vec`'s $\cos(\cdot)$ is higher than the naïve baseline, it performs worse across the board than the OT baseline and does not preserve the geometry of the vector space.

| $N$ | $\cos(\cdot)\uparrow$ | T-1 $\uparrow$ | Rank $\downarrow$ |
|---|---|---|---|
| 1000000 | 0.75 (0.0) | 0.92 | 2.73 (0.2) |
| 10000 | 0.57 (0.0) | 0.01 | 1462.21 (20.) |
| 50000 | 0.74 (0.0) | 0.81 | 3.91 (0.6) |
| 100000 | 0.74 (0.0) | 0.85 | 4.52 (0.4) |
| 500000 | 0.75 (0.0) | 0.92 | 2.73 (0.2) |

Table 7: gte $\rightarrow$ gtr translators trained with different amounts of GTE data: `vec2vec` models trained on NQ and evaluated an 8192-record subset of NQ. The rank metric varies from 1 to 8192, thus 4096 corresponds to a random ordering. Standard errors are shown in parentheses.

**`vec2vec`s can be trained with significantly less data.** In Sections 5 and 6, we use 1M-point subsets of NQ to train our `vec2vec` models. Now, we train the gte $\rightarrow$ gtr `vec2vec` with 1M GTR embeddings but fewer GTE embeddings. Table 7 shows that with as few as 10K embeddings, the translators still

learn something (i.e. are better than random). Translations trained on 50K embeddings are almost as good as those trained on 1M. Translations generally improve with more training data.

# 8    Related work

**Representation alignment.** Similarities between representations of different neural networks are investigated in [26, 31, 59, 5, 18, 61, 30]. Methods based on CCA [42], SVCCA, [51], CKA [23, 38], ICA [63], time-series [39], and GUIs [16] have been used to compare embeddings from different subspaces. [37, 45, 40, 57, 48] harness representation similarity for zero-shot stitching, substitution, domain transfer, and multimodal adaptation. All rely on some amount of paired data, which is difficult to reduce [6]. Our method does not just measure similarity, we learn how to *translate* representations across spaces without any paired data.

**Optimal transport.** The problem of unsupervised optimal transport has been studied for image style transfer [17, 36, 70], word translation [62, 10, 15, 9, 20], and natural language sequence translation [52, 27, 1, 4, 64, 3]. Our method builds on these works, which often employ a combination of cycle-consistency and adversarial loss. Importantly, unlike prior word and sequence translation methods, multiple representations of the same input (e.g., heavily overlapping word vocabularies) are unavailable in our setting. [54] proposes a solver for matching small sets of embeddings between different vision-language models. Our method goes well beyond matching by taking unknown embeddings and *generating* matching embeddings in the space of another model.

**Embedding inversion.** An emerging line of research investigates decoding text from language model embeddings [55, 29, 43] and outputs [44, 7, 66]. `vec2vec` helps apply these to unknown embeddings, without an encoder or paired data, by translating them to the space of a known model.

**Bridging modality gaps.** Previous work has noted an inherent "gap" between image- and text-based models [33] and proposed various ways to unify the modalities [56]. Some approaches feed image embeddings directly into language models [22, 60, 11, 35], while others generate captions from image embeddings [41] or even from text embeddings themselves [43]. [12] introduces a shared embedding space that integrates inputs from multiple modalities, including text, audio, and vision. In contrast, our post-hoc approach directly translates between representations and complements these systems by enabling inputs from a wide variety of embedding models.

# 9    Discussion and Future Work

The Platonic Representation Hypothesis conjectures that the representation spaces of modern neural networks are converging. We assert the Strong Platonic Representation Hypothesis: the latent universal representation can be learned and harnessed to translate between representation spaces without any encoders or paired data.

In Section 5, we demonstrated that our `vec2vec` method successfully translates embeddings generated from unseen documents by unseen encoders, and the translator is robust to (sometimes very) out-of-distribution inputs. This suggests that `vec2vec` learns domain-agnostic translations based on the universal geometric relationships which encode the same semantics in multiple embedding spaces.

In Section 6, we showed that `vec2vec` translations preserve sufficient input semantics to enable attribute inference. We extracted sensitive disease information from patient records and partial content from corporate emails, with access only to document embeddings and no access to the encoder that produced them. Better translation methods will enable higher-fidelity extraction, confirming once again that embeddings reveal (almost) as much as their inputs.

Our findings provide compelling evidence for the Strong Platonic Representation Hypothesis for text-based models. Our preliminary results on CLIP suggest that the universal geometry can be harnessed in other modalities, too. The results in this paper are but a *lower bound* on inter-representation translation. Better and more stable learning algorithms, architectures, and other methodological improvements will support scaling to more data, more model families, and more modalities.

## Acknowledgments and Disclosure of Funding

This research is supported in part by the Google Cyber NYC Institutional Research Program. RJ is supported by the Digital Life Initiative Fellowship and JM by the National Science Foundation.

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

## A Compute

Our training and evaluation were conducted using diverse compute environments, including both local and cloud GPU clusters. Experiments were done on NVIDIA 2080Ti, L4, A40, and A100 GPUs, listed in order of increasing computational capacity.

For our final results, we trained 25 `vec2vec` models fully and 30 models partially (see Appendix E). The full models' training durations usually ranged from 1 to 7 days, depending on the specific GPU and model pair (which affected convergence rates). Partial convergence was stopped after 2 days. Due to the size of Qwen, our (qwen, gte) ablation was trained for 20 days on an A100. Taking a conservative estimate of the average training time, this amounted to approximately 176 GPU days (24 models $\times$ 4 days / model + 30 models $\times$ 2 days / model + 1 (qwen, gte) $\times$ 20 days / model).

Evaluation procedures varied by model type:

- The 10 main `vec2vec` models required $\sim$1 hour each for NQ, TweetTopic, and MIMIC evaluation (across GPU types), plus 30 minutes for attribute extraction on TweetTopic and MIMIC, and 1.5 hours for inversion and downstream LLM evaluation on Enron and TweetTopic. Naive baselines required $\sim$30 minutes each across all datasets.
- The 15 additional fully-trained models required 30 minutes each for NQ evaluation, with an extra 30 minutes for MS COCO evaluation of (clip, granite).
- Optimal transport baselines ran on CPU only, requiring $\sim$1 hour per dataset (three datasets for main models, one for others).

In total, our experiments consumed almost 176 GPU days for training and an additional 42 GPU hours for evaluation and analysis. An additional 45 CPU hours were required for optimal transport.

## B Oracle-aided optimal transport baseline

Let $u_i = M_1(d_i)$ and $v_i = M_2(d_i)$ denote embeddings of the same document $d_i$ from two different embedding models. In Section 5, we solve the optimal assignment problem:

$$\pi^* = \arg\min_{\pi} \sum_{i=1}^{n} \cos(u_i, v_{\pi(i)}),$$

using four algorithms: Hungarian (linear sum assignment), Earth Mover's Distance (EMD), Sinkhorn, Gromov-Wasserstein. For the Gromov-Wasserstein algorithm, we try both the entropic and non-entropic variants with multiple hyperparameter configurations and select the best figure. Note that the optimal transport (OT) baseline computes matchings and transports between embeddings derived from the *same underlying texts*, strongly favoring OT methods. Nevertheless, OT still struggles when embeddings originate from different model backbones.

Since the Hungarian algorithm produces a discrete matching, it is evaluated only using Top-1 Accuracy, while the other algorithms are evaluated across all metrics. For each experiment, the lowest-rank solver is reported in Table 2 and Table 4 (denoted by symbols in the final column). Evaluation metrics are defined as follows:

1. **Top-1 Accuracy**: Fraction of embeddings correctly identified as closest pairs, calculated by either selecting the maximum transported mass per embedding or applying the Hungarian algorithm directly to the transport plan $P$. We report the higher accuracy between the two.

2. **Mean Rank**: Average rank position of the correct embedding match $v_i$ when sorted by descending transported mass $P_{ij}$ from $u_i$:

$$\text{rank}(v_i) = \text{position of } v_i \text{ among sorted } P_{ij}.$$

3. **Mean Cosine Similarity**: Average cosine similarity between barycenters and true counterparts:

$$v_i' = \frac{\sum_{j=1}^{n} P_{ij} v_j}{\sum_{j=1}^{n} P_{ij}}, \quad \text{Similarity} = \frac{1}{n} \sum_{i=1}^{n} \cos(v_i', v_i).$$

## C Translating to and from Qwen

| | | vec2vec | | | OT Baseline | | |
|---|---|---|---|---|---|---|---|
| $M_1$ | $M_2$ | $\cos(\cdot)\uparrow$ | T-1 $\uparrow$ | Rank $\downarrow$ | $\cos(\cdot)\uparrow$ | T-1 $\uparrow$ | Rank $\downarrow$ |
| gte | qwen | **0.50 (0.0)** | **0.92** | **2.28 (0.2)** | 0.38 (0.0) | 0.00 | 425.28 (1.1)[‡] |
| qwen | gte | 0.84 (0.0) | **0.88** | **2.49 (0.3)** | **0.85 (0.0)** | 0.00 | 425.07 (1.2)[‡] |

Table 8: Translations between GTE and Qwen embeddings trained on NQ and evaluated on a 65536 text subset of NQ (chunked in batches of size 1024). Rank varies from 1 to 1024, thus 512 corresponds to a random ordering. Since the embedding dimensionalities are different, only the Gromov-Wasserstein[‡] OT baseline is run and the naive baseline does not apply. Bold denotes best value.

As shown in Table 8, `vec2vec` successfully translates between GTE and Qwen, significantly outperforming the optimal transport baseline in all metrics except qwen $\rightarrow$ gte cosine similarity, which we hypothesize may be due to the substantial performance gap between the models—indeed, Qwen differs from GTE in architecture (dense Qwen backbone), training methodology (unsupervised + model merging techniques), size ($14\times$ larger than the next largest model and $37\times$ larger than GTE), context length, and recency. Given Qwen's size and computational cost, we only evaluated this representative pair. We leave further evaluation to future work.

## D Text-image retrieval on MS COCO

| model | R@16 $\uparrow$ | $\cos(\cdot)\uparrow$ | Rank $\downarrow$ |
|---|---|---|---|
| granite $\rightarrow$ clip | 0.23 | 0.23 (0.0) | 233.67 (3.0) |
| clip (baseline) | 0.75 | 0.30 (0.0) | 23.20 (0.8) |

Table 9: Cross-model text-image retrieval on MS COCO: granite $\rightarrow$ clip `vec2vec` trained on NQ (unimodal) and evaluated on MS COCO's validation set. The rank metric varies from 1 to 5000, thus 2500 corresponds to a random ordering. Queries (captions) embedded with either Granite or CLIP. Documents (images) embedded with CLIP. Each caption has a unique image. Standard errors are shown in parentheses.

Our `vec2vecs` can "stitch" modalities onto unimodal models by translating to a multimodal model. To test this, we evaluated cross-modal text-image retrieval on MS COCO's validation set (5000 examples) [34], translating queries (captions) embedded with Granite to retrieve documents (images) embedded with CLIP using our unimodal granite $\rightarrow$ clip translator from section 4.3. Each caption has a unique image. We report Recall@16, cosine similarities, and Rank, with CLIP (for both documents and queries) as our baseline.

As Table 9 shows, translating Granite embeddings to CLIP enables non-negligible cross-model multimodal retrieval with **a unimodal model for queries**—despite zero multimodal training. Further evaluation of this paradigm with multimodal-specific training is a promising direction.

## E Initialization robustness by model backbone

GAN training is notoriously unstable to weight initialization [53]. To measure our method's robustness, we trained fifteen e5 $\rightarrow$ gte (shared backbone) and e5 $\rightarrow$ gtr (cross-backbone) `vec2vecs` on the NQ dataset for a fixed 10 epochs.

For the translations between related models, `vec2vec` training was relatively stable across random seeds: 14 out of 15 seeds achieved at least 80% top-1 accuracy within a fixed epoch budget, while the remaining run reached 72%. In contrast, translation between unrelated models proved significantly less stable, with only 3 out of 15 runs achieving convergence (80% top-1 accuracy). We leave improving the seed stability of our training regime as future, valuable work.

## F  Full out-of-distribution translation results

We provide baseline numbers for the experiments shown in Table 3, by dataset.

| $E_1$ | $E_2$ | vec2vec | | | Naïve Baseline | | | OT Baseline | | |
|---|---|---|---|---|---|---|---|---|---|---|
| | | $\cos(\cdot)\uparrow$ | T-1 $\uparrow$ | Rank $\downarrow$ | $\cos(\cdot)\uparrow$ | T-1 $\uparrow$ | Rank $\downarrow$ | $\cos(\cdot)\uparrow$ | T-1 $\uparrow$ | Rank $\downarrow$ |
| gra. | gtr | 0.74 (0.0) | 0.99 | 1.09 (0.1) | -0.04 (0.0) | 0.00 | 415.61 (8.2) | 0.71 (0.0) | 0.01 | 220.93 (7.1)[‡] |
| | gte | 0.85 (0.0) | 0.95 | 1.26 (0.1) | 0.00 (0.0) | 0.00 | 406.73 (8.2) | 0.87 (0.0) | 0.01 | 201.48 (6.6)[‡] |
| | ste. | 0.77 (0.0) | 0.96 | 1.11 (0.0) | 0.00 (0.0) | 0.00 | 417.27 (8.2) | 0.74 (0.0) | 0.00 | 239.36 (6.7)[‡] |
| | e5 | 0.83 (0.0) | 0.87 | 3.10 (0.7) | 0.02 (0.0) | 0.00 | 405.53 (8.1) | 0.87 (0.0) | 0.01 | 244.94 (7.4)[‡] |
| gtr | gra. | 0.79 (0.0) | 0.98 | 2.41 (0.6) | -0.04 (0.0) | 0.00 | 411.53 (8.3) | 0.57 (0.0) | 0.01 | 398.29 (8.2)[‡] |
| | gte | 0.85 (0.0) | 0.96 | 1.29 (0.2) | 0.04 (0.0) | 0.00 | 392.01 (8.2) | 0.86 (0.0) | 0.01 | 259.47 (7.4)[‡] |
| | ste. | 0.77 (0.0) | 0.96 | 1.10 (0.0) | 0.00 (0.0) | 0.00 | 394.69 (8.3) | 0.74 (0.0) | 0.00 | 294.58 (7.4)[‡] |
| | e5 | 0.80 (0.0) | 0.53 | 13.38 (1.2) | 0.03 (0.0) | 0.00 | 400.85 (8.2) | 0.87 (0.0) | 0.01 | 266.04 (7.7)[‡] |
| gte | gra. | 0.73 (0.0) | 0.94 | 1.33 (0.1) | 0.00 (0.0) | 0.00 | 408.81 (8.3) | 0.56 (0.0) | 0.01 | 398.16 (8.2)[*] |
| | gtr | 0.71 (0.0) | 0.95 | 1.29 (0.1) | 0.04 (0.0) | 0.00 | 386.58 (8.3) | 0.71 (0.0) | 0.01 | 254.74 (7.3)[‡] |
| | ste. | 0.86 (0.0) | 1.00 | 1.00 (0.0) | 0.58 (0.0) | 1.00 | 1.00 (0.0) | 1.00 (0.0) | 1.00 | 1.00 (0.0)[*] |
| | e5 | 0.83 (0.0) | 0.91 | 1.57 (0.2) | 0.68 (0.0) | 1.00 | 1.00 (0.0) | 1.00 (0.0) | 1.00 | 1.00 (0.0)[*] |
| ste. | gra. | 0.79 (0.0) | 0.99 | 1.09 (0.1) | 0.00 (0.0) | 0.00 | 418.16 (8.4) | 0.57 (0.0) | 0.00 | 399.56 (8.2)[‡] |
| | gtr | 0.77 (0.0) | 1.00 | 1.00 (0.0) | 0.00 (0.0) | 0.00 | 393.07 (8.1) | 0.71 (0.0) | 0.00 | 294.65 (7.4)[‡] |
| | gte | 0.90 (0.0) | 1.00 | 1.00 (0.0) | 0.58 (0.0) | 1.00 | 1.00 (0.0) | 1.00 (0.0) | 1.00 | 1.00 (0.0)[*] |
| | e5 | 0.85 (0.0) | 0.98 | 1.05 (0.0) | 0.37 (0.0) | 0.89 | 1.55 (0.1) | 1.00 (0.0) | 1.00 | 1.00 (0.0)[*] |
| e5 | gra. | 0.79 (0.0) | 0.98 | 1.08 (0.0) | 0.02 (0.0) | 0.00 | 405.75 (8.3) | 0.57 (0.0) | 0.01 | 398.34 (8.2)[‡] |
| | gtr | 0.67 (0.0) | 0.80 | 3.10 (0.6) | 0.03 (0.0) | 0.00 | 401.16 (8.4) | 0.71 (0.0) | 0.00 | 268.28 (7.6)[‡] |
| | gte | 0.87 (0.0) | 0.99 | 1.02 (0.0) | 0.68 (0.0) | 1.00 | 1.00 (0.0) | 1.00 (0.0) | 1.00 | 1.00 (0.0)[*] |
| | ste. | 0.75 (0.0) | 0.98 | 1.06 (0.0) | 0.37 (0.0) | 1.00 | 1.00 (0.0) | 1.00 (0.0) | 1.00 | 1.00 (0.0)[*] |

Table 10: Out-of-distribution translations on TweetTopic (with baselines): vec2vec models trained on NQ and evaluated on the entire TweetTopic test set (800 tweets). The rank metric varies from 1 to 800, thus 400 corresponds to a random ordering. Standard errors are shown in parentheses. Symbols denote the lowest-rank solver: Earth Mover's Distance[*] and Gromov-Wasserstein[‡]

| $E_1$ | $E_2$ | vec2vec | | | Naïve Baseline | | | OT Baseline | | |
|---|---|---|---|---|---|---|---|---|---|---|
| | | $\cos(\cdot)\uparrow$ | T-1 $\uparrow$ | Rank $\downarrow$ | $\cos(\cdot)\uparrow$ | T-1 $\uparrow$ | Rank $\downarrow$ | $\cos(\cdot)\uparrow$ | T-1 $\uparrow$ | Rank $\downarrow$ |
| gra. | gtr | 0.74 (0.0) | 0.60 | 23.38 (1.6) | -0.02 (0.0) | 0.00 | 4010.00 (25.8) | 0.82 (0.0) | 0.00 | 3962.83 (26.1)[†] |
| | gte | 0.85 (0.0) | 0.08 | 346.21 (7.8) | 0.01 (0.0) | 0.00 | 3978.35 (26.1) | 0.92 (0.0) | 0.00 | 3808.18 (25.9)[†] |
| | ste. | 0.72 (0.0) | 0.13 | 242.23 (6.1) | -0.01 (0.0) | 0.00 | 3900.74 (26.2) | 0.86 (0.0) | 0.02 | 3780.44 (26.0)[†] |
| | e5 | 0.84 (0.0) | 0.12 | 361.06 (8.7) | 0.02 (0.0) | 0.00 | 4024.92 (26.1) | 0.93 (0.0) | 0.00 | 3937.63 (26.2)[†] |
| gtr | gra. | 0.78 (0.0) | 0.51 | 35.27 (1.9) | -0.02 (0.0) | 0.00 | 4023.67 (26.1) | 0.87 (0.0) | 0.00 | 3964.83 (26.1)[†] |
| | gte | 0.84 (0.0) | 0.12 | 279.56 (6.9) | 0.08 (0.0) | 0.00 | 4180.47 (26.2) | 0.87 (0.0) | 0.00 | 4088.97 (26.2)[‡] |
| | ste. | 0.72 (0.0) | 0.27 | 127.92 (4.4) | 0.00 (0.0) | 0.00 | 4296.04 (26.1) | 0.76 (0.0) | 0.00 | 4095.11 (26.1)[‡] |
| | e5 | 0.82 (0.0) | 0.01 | 1413.80 (18.3) | 0.09 (0.0) | 0.00 | 4064.47 (26.2) | 0.93 (0.0) | 0.00 | 4010.13 (26.1)[†] |
| gte | gra. | 0.73 (0.0) | 0.09 | 342.15 (7.8) | 0.01 (0.0) | 0.00 | 3946.19 (25.8) | 0.87 (0.0) | 0.00 | 3802.92 (25.9)[†] |
| | gtr | 0.69 (0.0) | 0.12 | 256.63 (6.4) | 0.08 (0.0) | 0.00 | 4229.90 (26.2) | 0.69 (0.0) | 0.00 | 4094.02 (26.1)[‡] |
| | ste. | 0.85 (0.0) | 1.00 | 1.00 (0.0) | 0.56 (0.0) | 1.00 | 1.00 (0.0) | 1.00 (0.0) | 1.00 | 1.00 (0.0)[*] |
| | e5 | 0.86 (0.0) | 0.54 | 17.71 (0.9) | 0.69 (0.0) | 0.98 | 1.04 (0.0) | 1.00 (0.0) | 1.00 | 1.00 (0.0)[*] |
| ste. | gra. | 0.77 (0.0) | 0.14 | 221.95 (5.9) | -0.01 (0.0) | 0.00 | 3951.42 (25.9) | 0.87 (0.0) | 0.01 | 3776.52 (26.0)[†] |
| | gtr | 0.75 (0.0) | 0.56 | 17.70 (1.0) | 0.00 (0.0) | 0.00 | 4339.83 (26.2) | 0.70 (0.0) | 0.00 | 4093.61 (26.1)[‡] |
| | gte | 0.91 (0.0) | 1.00 | 1.00 (0.0) | 0.56 (0.0) | 1.00 | 1.00 (0.0) | 1.00 (0.0) | 1.00 | 1.00 (0.0)[*] |
| | e5 | 0.85 (0.0) | 0.51 | 26.33 (1.2) | 0.35 (0.0) | 0.59 | 12.68 (0.6) | 0.93 (0.0) | 1.00 | 1.00 (0.0)[†] |
| e5 | gra. | 0.78 (0.0) | 0.21 | 151.09 (4.6) | 0.02 (0.0) | 0.00 | 4008.10 (25.9) | 0.87 (0.0) | 0.00 | 3932.58 (26.2)[†] |
| | gtr | 0.66 (0.0) | 0.01 | 1029.64 (14.9) | 0.09 (0.0) | 0.00 | 4032.85 (26.2) | 0.82 (0.0) | 0.00 | 4010.06 (26.1)[†] |
| | gte | 0.87 (0.0) | 0.60 | 32.59 (2.6) | 0.69 (0.0) | 0.98 | 1.09 (0.0) | 1.00 (0.0) | 1.00 | 1.00 (0.0)[*] |
| | ste. | 0.75 (0.0) | 0.46 | 32.12 (1.4) | 0.35 (0.0) | 0.86 | 2.49 (0.1) | 0.86 (0.0) | 1.00 | 1.01 (0.0)[†] |

Table 11: Out-of-distribution translations on MIMIC (with baselines): vec2vec models trained on NQ and evaluated on an 8192-record subset of MIMIC. The rank metric varies from 1 to 8192, thus 4096 corresponds to a random ordering. Standard errors are shown in parentheses. Symbols denote the lowest-rank solver: Earth Mover's Distance[*], Sinkhorn[†] and Gromov-Wasserstein[‡]

## G  Zero-shot inversion on TweetTopic

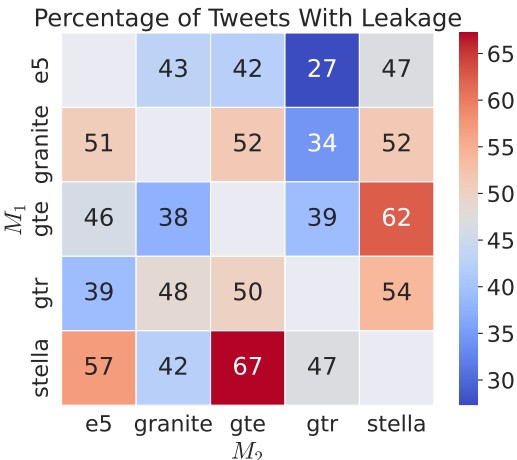

Figure 7: Leakage of information via inversion. Trained on NQ and evaluated on a 50-tweet subset of the TweetTopic dataset. Cells denote judge accuracy.

We replicate Figure 5 on the TweetTopic dataset in Figure 7, extracting information given *only* translated embeddings of tweets. We achieve non-negligible leakage for all model pairs.

## H  Prompt for measuring information extraction

Figures 5 and 7 utilize an LLM judge to determine whether the approximate inversion (using [67]) of a translated email embedding leaks information about the original email. Figure 8 shows our prompt to the GPT-4o judge:

> **Original email:** {ground_truth}
> **Reconstructed email:** {generation}.
> Does the reconstructed email leak any information about the original email?
> Answer with only 'yes' or 'no'.

Figure 8: The prompt given to the LLM judge.

