# OpenReview forum: "Harnessing the Universal Geometry of Embeddings"
_NeurIPS.cc/2025/Conference — NeurIPS 2025 poster_

### Official Review · Reviewer_qFGw · 2025-07-02

**Clarity:** 4
**Significance:** 3
**Originality:** 4
**Rating:** 5
**Confidence:** 4

**Summary:**

This paper learns how to build a mapping from a distribution of data encoded in one embedded space into another space (of the same dimension) where the encoding is know.  It is assumed that the distributions of the two datasets are known, and then by aligning the distributions with GAN-style loss function one can understand the meaning of embedded items in the first data set.

Empirically, the paper shows that if the architectures have a lot of commonalities (same architecture, embedded elements random from same data set), then the method works quite well.  If not, the performance degrades some, but is still pretty good.

**Questions:**

I would like to understand how well this particular method worked against baselines.  Or why we could not compare against those.

**Ethical Concerns:**

["NO or VERY MINOR ethics concerns only"]

**Final Justification:**

I am convinced by the author's rebuttal and new experiments.  I think this is an important paper, that provides interesting insights into what is captured in language models, and will certainly spawn more work trying to replicated and improve upon their results.

**Limitations:**

yes.

**Quality:**

3

**Strengths And Weaknesses:**

Strengths:
  + innovative approach, and provides some new evidence for the Plantonic Representation Hypothesis.

  + I was surprised to see how well it worked on datasets that shared a common backbone (although naive methods already work well), or hidden test was randomly split from test set.

  + I like the experimental design.  It tested a variety of things, on several architectures, evaluations/data sets, and types of data (text/image).

  + The paper is well written.


Weaknesses:
  - There seems nothing specific to LLMs here, as opposed to work from 5-10 years ago that applied translation to non-contextual language models (e.g., word2vec, GloVe) for different languages. These ideas (which seem natural) could have been applied there.  This decreases the novelty to me.

  - I would have liked to see a direct comparison against translation of language models (e.g., Conneau etal 2018) as a better baseline.  What is gained with the multi-term GAN loss function in this paper.

  - The paper does not explain how it does alignment in full detail.  It says "via either the Hungarian, Earth Mover’s Distance, Sinkhorn, or Gromov-Wasserstein algorithms, choosing the best-performing solver per experiment" which is quite un-satisfying.

---

> ### Author Rebuttal · Authors · 2025-07-31
>
> Thank you for your response.  We respond point by point.
>
> ---
>
> **(W1):** Methods for unsupervised transport on word embeddings do not solve our problem because they rely on heavily-overlapping word vocabularies between languages. For example, when translating from English to Spanish, ‘cat’ and ‘gato’ appear in both the source and target training data, respectively.
>
> Our setting is different and significantly harder.  We work with embeddings of relatively long text sequences.  Given the embedding of an unknown sequence in the source space, it is not guaranteed that the training set contains an embedding of the same or even a related sequence in the target space.  This makes our assignment problem–that of mapping entire documents, not just individual words–much more difficult.
>
> Tables 2, 4, 6, 7 show that even when we use optimal assignment methods to compute translations between embeddings derived from **the same underlying texts** (we call this “oracle-aided”), they struggle to produce good assignments for embeddings that originate from different model backbones.
>
> We have rewritten the introduction to clarify this distinction.
>
> ---
>
> **(W2, Q1):** Conneau et al’s method is similar at a very high level (they also define the problem as adversarial learning) but it does not apply directly in our setting for the following reasons:
>
> 1. Our non-overlapping setting (i.e., the inputs for the source and target embeddings during GAN training are not the same) breaks the key assumption behind the Procrustes step in their method.
> 2. Since we don’t have access to the embeddings’ underlying text, refinement methods based on frequency proposed by Conneau et al cannot be used.
>
> We do compare our approach with word translation methods that potentially apply in our setting.  For example, [1] requires neither overlapping supports, nor frequency-based analysis.
>
> Tables 2, 4, 6, 7 in our paper measure the performance of OT methods, including [1].  Even with perfectly overlapping supports (an unrealistic assumption which favors OT), **OT works only for a few encoder pairs and fails when the encoders are from different model families.** For example, when translating from GTE to Stella on NQ (both BERT-based), the best OT method (in this case Sinkhorn) achieves a perfect rank and accuracy of 1.0 and 1.0. However, when translating from GTR to Stella, [1] (the best performing method for this pair) achieves only a near-random rank of 4095.46 and accuracy of 0.0. These numbers improve slightly (rank of 3559.06 and accuracy of 0.0) by introducing entropic regularization to [1], but they still fail for embedding translation and significantly underperform vec2vec.
>
> In the next version of the paper, we will clarify the above and expand our comparison with entropic Gromov-Wasserstein once it concludes this weekend (results below).
>
> [1]  Alvarez-Melis, et al. “Gromov-Wasserstein Alignment of Word Embedding Spaces”. 2018.
>
> ---
>
> **(W2):** For our GAN implementation, Appendix C demonstrates that **each term is critical.** When translating from GTE to GTR on NQ, vec2vec achieves an average rank of 2.64. If any one component of our method is removed (VSP loss, CC loss, latent GAN loss, or hyperparameter tuning), performance drops close to random guessing, with ranks of 4196.64, 3941.36, 3897.09, and 4011.73, respectively. Furthermore, cosine similarities plummet from 0.75 to ~0.5, and accuracies drop from 91% to 0%.
>
> We will highlight this discussion in the paper.
>
> ---
>
> **(W3):** Sorry about this. We have added a section to the Appendix detailing these optimal transport (OT) experiments, covering the methods used, evaluation metrics, and experimental setup. Additionally, as illustrated below for the NQ dataset, we now denote the best-performing OT model using symbols (Earth Mover’s Distance: ∗, Sinkhorn: †, and Gromov-Wasserstein: ‡), and report the average cosine similarity between barycenters and their true counterparts.
>
> We will further update this section once our evaluation of another baseline, entropic Gromov-Wasserstein, concludes this weekend.
>
> ---
>
> ### Updated Tables
> ```
> +--------+--------+----------+--------+--------------+
> |             Updated OT Baseline Table              |
> +--------+--------+----------+--------+--------------+
> |   M₁   |   M₂   | cos(·) ↑ | T-1 ↑  | Rank ↓       |
> +--------+--------+----------+--------+--------------+
> | gra.   | gtr    | 0.50(0.0)| 0.00   |4094.22(9.2)‡ |
> |        | gte    | 0.85(0.0)| 0.00   |4069.91(9.3)‡ |
> |        | ste.   | 0.45(0.0)| 0.00   |4096.35(9.2)‡ |
> |        | e5     | 0.68(0.0)| 0.00   |4096.17(9.2)‡ |
> +--------+--------+----------+--------+--------------+
> | gtr    | gra.   | 0.50(0.0)| 0.00   |4093.55(9.2)‡ |
> |        | gte    | 0.85(0.0)| 0.00   |4079.92(9.2)‡ |
> |        | ste.   | 0.46(0.0)| 0.00   |4093.85(9.0)‡ |
> |        | e5     | 0.83(0.0)| 0.00   |4066.42(9.2)‡ |
> +--------+--------+----------+--------+--------------+
> | gte    | gra.   | 0.69(0.0)| 0.00   |4069.23(9.2)‡ |
> |        | gtr    | 0.70(0.0)| 0.00   |4078.45(9.0)‡ |
> |        | ste.   | 0.69(0.0)| 1.00   |1.00(0.0)†    |
> |        | e5     | 0.83(0.0)| 1.00   |1.00(0.0)†    |
> +--------+--------+----------+--------+--------------+
> | ste.   | gra.   | 0.48(0.0)| 0.00   |4096.38(9.2)‡ |
> |        | gtr    | 0.50(0.0)| 0.00   |4095.46(9.3)‡ |
> |        | gte    | 0.86(0.0)| 1.00   |1.00(0.0)†    |
> |        | e5     | 0.83(0.0)| 1.00   |1.00(0.0)†    |
> +--------+--------+----------+--------+--------------+
> | e5     | gra.   | 0.69(0.0)| 0.00   |4096.12(9.2)‡ |
> |        | gtr    | 0.70(0.0)| 0.00   |4065.74(9.2)‡ |
> |        | gte    | 0.86(0.0)| 1.00   |1.00(0.0)†    |
> |        | ste.   | 0.68(0.0)| 1.00   |1.00(0.0)†    |
> +--------+--------+----------+--------+--------------+
> ```
>
> ```
> +--------+--------+----------+--------+--------------+
> |            Entropic Gromov-Wasserstein             |
> +--------+--------+----------+--------+--------------+
> |   M₁   |   M₂   | cos(·) ↑ | T-1 ↑  | Rank ↓       |
> +--------+--------+----------+--------+--------------+
> | gte    | gra.   | 0.69(0.0)| 0.00   |2664.38(8.6)‡ |
> |        | gtr    | 0.70(0.0)| 0.00   |3064.16(8.9)‡ |
> |        | ste.   | 0.67(0.0)| 0.00   |3457.99(9.0)‡ |
> |        | e5     | 0.83(0.0)| 0.00   |3096.15(8.5)‡ |
> +--------+--------+----------+--------+--------------+
> | gtr    | gra.   | 0.70(0.0)| 0.00   |2775.17(8.6)‡ |
> |        | gte    | 0.85(0.0)| 0.00   |3070.69(8.9)‡ |
> |        | ste.   | 0.67(0.0)| 0.00   |3559.06(9.1)‡ |
> |        | e5     | 0.83(0.0)| 0.00   |3888.01(8.9)‡ |
> +--------+--------+----------+--------+--------------+
> ```

---

> > ### Comment · Reviewer_qFGw · 2025-08-03
> >
> > Thanks for the rebuttal.  I have increased my score.  This is really thought-provoking work.

---

### Official Review · Reviewer_LNxu · 2025-07-03

**Clarity:** 4
**Significance:** 4
**Originality:** 3
**Rating:** 5
**Confidence:** 4

**Summary:**

The main contribution of the paper is a method (named "vec2vec") for learning model-agnostic vector representations of text documents. This leans on a representation hypothesis which states that given sufficiently many data and model capacity, there should be a common latent representation. Further, the authors conjecture that for text this is also possible without enforced alignment of known pairs of data from different representations. Using mappings that connect vector representations learned by different models, the authors leverage those model-agnostic representations as a "bridge" or "universal representation". This is achieved assuming only access to a learned set of vector representations of some unknown text data and then a known data set.

**Questions:**

1. Given how the chosen "oracle-aided optimal assignment" performed in the experiments sometimes abysmally, it made me wonder why that was the case. It seems the paper did not address that, at least per experiment. Could you clarify this?
2. Following (1), could that be an indicator that this is not an appropriate baseline or another "upper bound" baseline could be adequate?
3. Could the authors clarify how often does a training run need to be restarted to achieve a converged optimisation procedure?

**Ethical Concerns:**

["NO or VERY MINOR ethics concerns only"]

**Final Justification:**

Despite its pragmatic nature, the paper is experimentally thorough and shows itself to be a good first attempt at solving a tricky problem.

While initially the authors wrote the paper in an optimistic tone, if they adopt the changes suggested by me and other reviewers, including clarifying stability (or lack thereof) and compute requirements, I am satisfied with the methodology adopted and recommend acceptance.

**Limitations:**

The major limitation I see is caused by the lack of stable training. I appreciate that the authors also discussed direct societal impacts of their work, in particular to data leakage in the form of vector representations, being more serious than previously thought.

**Paper Formatting Concerns:**

- Table numbers and titles should appear before the table (all tables need to be fixed).

**Quality:**

3

**Strengths And Weaknesses:**

**Strengths**

Overall, the paper is clearly written, properly motivated, and the authors place it well amongst existing literature. Technically, the methodological build-up is easy to follow and intuitive. Importantly, the authors highlight the possibility of information extraction from leaked vector databases, which is critical from an information security standpoint.

**Weaknesses**

While intuitive, the methodology adopted is quite pragmatic, having a number of hyperparameters in the loss and architectural possibilities. As a concrete example of the pragmatism adopted, the authors forego investigation of instability in their proposed methodology, instead opting for investing more compute resources to achieve the desired outcomes. While I think that dedicating too much of the paper on making training stable could distract from the main point of the paper, it still passes an impression of being preliminary work. As I see it, it becomes one of the weakest points of the adopted methodology: the unstable training is rooted in the choices made that simplify the treatment of the problem as a more intuitive adversarial objective.

As a side note, I believe the chosen title is somewhat misleading, as there is no geometric treatment of the problem setting addressed in the paper. I would recommend the authors either clarify why this terminology is chosen at certain points (e.g., l. 56) in the paper or move away from it. As a suggestion, arguing in this direction using cosine similarities as evidence, you could consider using "representation" or "vector alignment" instead.

---

> ### Author Rebuttal · Authors · 2025-07-31
>
> Thanks for your review–we’ll respond point-by-point below:
>
> ---
> **(W1--Part 1):** While we agree that our method and framing are complex, our ablations (in Appendix C) show that **each part of our methodology appears necessary to learn translations** (that work better than random guessing)**.**  When translating from GTE to GTR on NQ, vec2vec achieves an average rank of 2.64. If any one component of our method is removed (VSP loss, CC loss, latent GAN loss, or hyperparameter tuning), performance drops close to random guessing, with ranks of 4196.64, 3941.36, 3897.09, and 4011.73, respectively. Furthermore, cosine similarities plummet from 0.75 to ~0.5, and accuracies drop from 91% to 0%.
>
> As demonstrated by the failure of oracle-aided optimal assignment baselines, the problem is hard.  Prior to our work, it was not known or even expected that a black-box, unsupervised, no-encoder translation between text embeddings can be learned at all.  Our work provides the first constructive proof that such a method exists, and we hope that future research will improve it.
>
> ---
> **(W1--Part 2):** We found that the adversarial setup was necessary to bridge the distance between the source and target embedding distributions in an unsupervised way. This formulation is standard in the literature [1, 2]. We experimented with other methods (such as iterative diffusion and normalizing flows) but they did not succeed at learning translations.
>
> [1] Conneau, et al. “Word Translation Without Parallel Data”. 2018.
>
> [2] Huang, et al. “Multimodal Unsupervised Image-to-Image Translation”. 2018.
>
> ---
> **(W2):** Thank you for the note. Our use of the word “geometry” comes from the fact that semantic similarity in embedding spaces is typically measured via geometric angles between embedding vectors.  In linguistics, “geometry” is used to describe the shape or structure of semantics, i.e., how words and phrases are related).
>
> ---
> **(Q1, Q2):** We will add the following discussion to our paper. During our evaluation, we were also surprised that oracle-aided optimal assignment did not work well for all model choices. A few things to note:
>
> 1. The baselines do work when the underlying encoder models are from the same family (in our experiments, on any pairing of e5, gte, and stella models).
> 2. Even though these baselines support unsupervised assignment of **single-word** embeddings, they are not designed for non-overlapping sets of longer text sequences.
>
>    When each embedding represents a single word, support sets for any large set of embedding vectors contain mostly the same semantic information (in different languages). For example, when translating from English to Spanish, the embeddings for ‘cat’ and ‘gato’ appear in the source and target distributions, respectively.
>
>    Our setting is different and significantly harder.  We work with embeddings of relatively long text sequences.  Given an embedding vector of an unknown sequence in the source embedding space, it is not guaranteed that the training set contains an embedding of the same or even closely related sequence in the target space.  This makes the assignment problem much more difficult.  Whereas for most words, the set of embeddings from the target space already contains an assignment target, this is not the case for text sequences.
>
> 3. Better baselines: our work is the first to show that black-box translation is possible for text sequence embeddings in the non-overlapping, unsupervised regime.  Since there is no prior work for this setting, there are no natural baselines.
>
>    Instead, we use methods from the word translation literature as baselines. We include them in the Optimal Transport / Assignment baseline. We have added a section to the Appendix detailing these optimal transport (OT) experiments, covering the methods used, evaluation metrics, and experimental setup. Additionally, as illustrated below for the NQ dataset, we now denote the best-performing OT model using symbols (Earth Mover’s Distance: ∗, Sinkhorn: †, and Gromov-Wasserstein: ‡), and report the average cosine similarity between barycenters and their true counterparts.
>
>    We will further update this section once our evaluation of another baseline, entropic Gromov-Wasserstein, concludes this weekend (results below).
>
> ---
>
> **(Q3):** Based on reviewer feedback, we analyzed the impact of initialization on convergence and found that translations between related models (e.g., BERT-backbone) are stable across random seeds. When translating embeddings from E5 to GTE, 14 out of 15 seeds ‘converged,’ achieving at least 80% top-1 accuracy within a fixed epoch budget; the remaining run reached >70%. In contrast, translation between unrelated models, such as from E5 to GTR (cross-backbone), proved significantly less stable, with only 3 out of 15 runs achieving convergence. We are currently trying to make this regime more stable, and in future work we will certainly revisit the problem. In the meantime, we argue that showing the first evidence that these solutions exist is an important contribution. This analysis will be added to the appendix of our paper.
>
> ---
> ### Updated Tables
> ```
> +--------+--------+----------+--------+--------------+
> |             Updated OT Baseline Table              |
> +--------+--------+----------+--------+--------------+
> |   M₁   |   M₂   | cos(·) ↑ | T-1 ↑  | Rank ↓       |
> +--------+--------+----------+--------+--------------+
> | gra.   | gtr    | 0.50(0.0)| 0.00   |4094.22(9.2)‡ |
> |        | gte    | 0.85(0.0)| 0.00   |4069.91(9.3)‡ |
> |        | ste.   | 0.45(0.0)| 0.00   |4096.35(9.2)‡ |
> |        | e5     | 0.68(0.0)| 0.00   |4096.17(9.2)‡ |
> +--------+--------+----------+--------+--------------+
> | gtr    | gra.   | 0.50(0.0)| 0.00   |4093.55(9.2)‡ |
> |        | gte    | 0.85(0.0)| 0.00   |4079.92(9.2)‡ |
> |        | ste.   | 0.46(0.0)| 0.00   |4093.85(9.0)‡ |
> |        | e5     | 0.83(0.0)| 0.00   |4066.42(9.2)‡ |
> +--------+--------+----------+--------+--------------+
> | gte    | gra.   | 0.69(0.0)| 0.00   |4069.23(9.2)‡ |
> |        | gtr    | 0.70(0.0)| 0.00   |4078.45(9.0)‡ |
> |        | ste.   | 0.69(0.0)| 1.00   |1.00(0.0)†    |
> |        | e5     | 0.83(0.0)| 1.00   |1.00(0.0)†    |
> +--------+--------+----------+--------+--------------+
> | ste.   | gra.   | 0.48(0.0)| 0.00   |4096.38(9.2)‡ |
> |        | gtr    | 0.50(0.0)| 0.00   |4095.46(9.3)‡ |
> |        | gte    | 0.86(0.0)| 1.00   |1.00(0.0)†    |
> |        | e5     | 0.83(0.0)| 1.00   |1.00(0.0)†    |
> +--------+--------+----------+--------+--------------+
> | e5     | gra.   | 0.69(0.0)| 0.00   |4096.12(9.2)‡ |
> |        | gtr    | 0.70(0.0)| 0.00   |4065.74(9.2)‡ |
> |        | gte    | 0.86(0.0)| 1.00   |1.00(0.0)†    |
> |        | ste.   | 0.68(0.0)| 1.00   |1.00(0.0)†    |
> +--------+--------+----------+--------+--------------+
> ```
>
> ```
> +--------+--------+----------+--------+--------------+
> |            Entropic Gromov-Wasserstein             |
> +--------+--------+----------+--------+--------------+
> |   M₁   |   M₂   | cos(·) ↑ | T-1 ↑  | Rank ↓       |
> +--------+--------+----------+--------+--------------+
> | gte    | gra.   | 0.69(0.0)| 0.00   |2664.38(8.6)‡ |
> |        | gtr    | 0.70(0.0)| 0.00   |3064.16(8.9)‡ |
> |        | ste.   | 0.67(0.0)| 0.00   |3457.99(9.0)‡ |
> |        | e5     | 0.83(0.0)| 0.00   |3096.15(8.5)‡ |
> +--------+--------+----------+--------+--------------+
> | gtr    | gra.   | 0.70(0.0)| 0.00   |2775.17(8.6)‡ |
> |        | gte    | 0.85(0.0)| 0.00   |3070.69(8.9)‡ |
> |        | ste.   | 0.67(0.0)| 0.00   |3559.06(9.1)‡ |
> |        | e5     | 0.83(0.0)| 0.00   |3888.01(8.9)‡ |
> +--------+--------+----------+--------+--------------+
> ```

---

> > ### Comment · Reviewer_LNxu · 2025-08-04
> >
> > I thank and commend the authors for their efforts in the rebuttal. I am satisfied with the answers provided and will therefore increase my score.

---

> ### Author Response · Authors · 2025-08-07
> **Thanks for the response**
>
> Thank you very much for taking the time to respond to our rebuttal, and for raising your score! Please follow-up if you have any further questions.

---

### Official Review · Reviewer_vYwp · 2025-07-04

**Clarity:** 2
**Significance:** 2
**Originality:** 2
**Rating:** 4
**Confidence:** 3

**Summary:**

This work proposes a GAN-based unsupervised approach to translate embedding space of different text encoders to an aligned latent embedding space. The method does not rely on parallel data points, and only relies on embeddings encoded by different text embedding models. Similar to the Platonic Representation Hypothesis paper, the paper reveals the similar semantic structures that different text encoders learn to model. The paper also briefly shows that embedding inversion models can zero-shot apply on translated embeddings to inverse texts from embeddings, which show certain safety-related implications.

**Questions:**

See above.

**Ethical Concerns:**

["NO or VERY MINOR ethics concerns only"]

**Final Justification:**

As mentioned in the original review comments, there are weaknesses in the current version regarding unclear narrative (whether the focus of the paper is the hypothesis or the method), unclear motivation (why assuming access to paired data points from different encoders are unrealistic), certain overclaims (claimed novelty despite similar to CycleGAN), and unclear experiment settings in 4.1 other than briefly mentioning GAN is unstable. In the rebuttal, the authors partly addressed these concerns and promised to clarify these unclear descriptions (main narrative/motivation, experiment settings, contextualize the overclaims) in the next version. The score has been raised to 4 to reflect authors' efforts in the rebuttal.

**Limitations:**

See above.

**Quality:**

3

**Strengths And Weaknesses:**

Strengths:
1. Different from approaches that require parallel data points to align embedding space, such as those in multilingual and translation fields, etc., the proposed method does not require parallel data points.
2. The behavior of the method is quite comprehensively evaluated, on different tasks (including whether the translated embedding can retrieve the paired data point encoded by the target encoder; and on attribute inference and embedding to text inversion) and on texts of OOD domains.

Weaknesses:
1. From the current narrative of the abstract and introduction, it is not immediately clear what's the motivation of the proposed method, and whether the focus of the work is to reveal universal semantic structures learned by different encoders (``The strong Platonic Representation Hypothesis'') or proposing a method. If the former, the proposed hypothesis seems under-theorized to distinguish from the original Platonic Representation Hypothesis, and if the latter, the motivation for the method is a bit vague (see point 2).
2. It is not very well-justified why expecting embeddings of parallel data is unrealistic (line 47) and what is "our setting" - it seems like not much relevant description is provided before this section. For most cases, it is quite realistic to encode a few hundred texts with two encoders and learn even a simple linear projection to align them, even if one or both of them are API models. It might help to justify the setting more from the safety aspect, e.g., cases of vector base breach that risks text inversion, and put more emphasis on the text inversion experiments in the paper.
3. Experiment details (model details, hyperparameters, compute requirements etc) are not at all described in 4.1 other than briefly mentioning that GAN is not stable.
4. Certain overclaims exist in the paper including the framing of the method even though it's in spirit very similar to methods like CycleGAN.

---

> ### Author Rebuttal · Authors · 2025-07-31
>
> Thank you for the constructive review. We address it point by point, below.
>
> ---
> **(W1):** Thank you for pointing out that the abstract and introduction are confusing.  We will clarify in the next version that we are proposing **both a hypothesis and a method.**
>
> The original Platonic Representation Hypothesis (PRH) conjectures that all image models of sufficient size converge to the same latent representations. This version of PRH is not constructive, it does not yield a working method for **pure representation translation**: given a representation of one model (and no other information!), translate it into a semantically equivalent representation of another model.
>
> Our work extends the PRH constructively to text models.  We propose and establish empirically not only that universal latent representations exist, but also that they can be explicitly characterized. We give the first method that operates exclusively on the embedding vectors of large text sequences and translates them from one vector space to another, while preserving a significant amount of semantic information about the underlying inputs.
>
> ---
>
> **(W2):** Thank you for pointing out that our wording is unclear.  To clarify, the existence of a universal latent representation and the constructive ability to translate embeddings to and from this representation without any additional information and no access to the encoder are important from both the fundamental perspective and the security perspective. \
>  \
> Fundamental perspective: we aim to answer a basic question, **how much information about a text input is preserved by its embedding representation?** In other words, how much information is contained in the vector itself, with no access to the encoder, no paired data, no candidate inputs, no parallels in another vector space, and no frequency statistics?  We believe that this is a fundamentally interesting and important question regardless of the application setting. \
>  \
> Security perspective: when a vector database is compromised or leaked, often the adversary gains access only to the raw contents of the database, i.e., embedding vectors.  While in some settings auxiliary information may be available, this is not guaranteed.  Our results establish a basic lower bound: any vector database compromise will leak at least as much information as can be inferred by translation into a known embedding space + zero-shot inversion.  As we show, this already includes a serious amount of sensitive information. \
>  \
> We will clarify this in the next version of the paper.
>
> ---
>
> **(W3):** Sorry for the confusion. Model details are provided at the end of Section 3.1. Compute requirements are described in detail in Appendix A. We adapt CycleGAN to function on embeddings instead of images by replacing CNNs with residual MLPs. We additionally utilize layer normalization and SiLU nonlinearities. A single vec2vec training run takes anywhere from 1 to 7 days to train on NVIDIA 2080Tis, L4s, A40s, and A100s.
>
> All of our code is published with data, hyperparameters, and instructions on how to recreate each experiment. We will clarify this in the main text and add additional hyperparameters to the Appendix.
>
> We will also add analysis on the stability of our method. Based on reviewer feedback, we analyzed the impact of initialization on convergence and found that translations between related models (e.g., BERT-backbone) are stable across random seeds. When translating embeddings from E5 to GTE, 14 out of 15 seeds converged, achieving at least 80% top-1 accuracy within a fixed epoch budget. The remaining run reached >70%.
>
> In contrast, translation between unrelated models, such as from E5 to GTR (cross-backbone), proved significantly less stable, with only 3 out of 15 runs achieving convergence. We are currently working to make this regime more stable, and in future work plan to revisit the problem. In the meantime, we argue that showing the first evidence that these solutions exist is an important contribution.
>
> ---
>
> **(W4--Part 1):** Thank you for the comment.  We are committed to making sure each of our claims is backed up. To that end, we would be happy to contextualize any specific claims.
>
> In our own review of the submitted version of the paper, we identified this claim “vec2vec is the first method to successfully translate embeddings from the space of one model to another without any paired data” as potentially too broad, and modified to provide the following context: “Prior work has successfully translated word embeddings between languages, typically relying on overlapping vocabularies across languages. In contrast, we translate embeddings of entire sequences between model spaces.“
>
> ---
> **(W4--Part 2):** While based on CycleGAN, our work is the first to use this method for pure representation translation, i.e., to learn a universal representation for translating embeddings of entire text sequences between model spaces **without encoder access or paired data**.
>
> Previously, CycleGAN was used for style transfer in images [1] and standard GANs were used to translate word embeddings between languages, typically relying on overlapping vocabularies across languages [2].  As we demonstrate in Appendix C, CycleGAN does not work out of the box. To create vec2vec, we apply an additional GAN on the latent space and introduce an extra loss term (VSP) for the first time. Our ablations in the appendix show that both are critical for pure representation translation. When translating from GTE to GTR on NQ, vec2vec achieves an average rank of 2.64. If any one component of our method is removed (VSP loss, CC loss, latent GAN loss, or hyperparameter tuning), performance drops close to random guessing, with ranks of 4196.64, 3941.36, 3897.09, and 4011.73, respectively. Furthermore, cosine similarities plummet from 0.75 to ~0.5, and accuracies drop from 91% to 0%.
>
> [1] Huang, et al. “Multimodal Unsupervised Image-to-Image Translation”. 2018.
>
> [2] Conneau, et al. “Word Translation Without Parallel Data”. 2018.

---

### Official Review · Reviewer_5c82 · 2025-07-05

**Clarity:** 3
**Significance:** 1
**Originality:** 3
**Rating:** 4
**Confidence:** 3

**Summary:**

The authors point out that the embedding spaces of current text encoders tend to converge. Based on this assumption, they propose a novel method called vec2vec, which transforms the text embeddings from one model to another. This is a completely new unsupervised approach that does not require paired data—only the vector embeddings from the source model and access to the target model are needed. The authors decompose the embedding translation task into three components: adapter, shared backbone, and adapter. And they employ the idea of cycle-GAN for training. The proposed method achieves promising results on the embedding translation task.

**Questions:**

1.	The authors need to improve the zero-shot inversion component and provide a reasonable evaluation of the reconstruction percentage on datasets such as TweetTopic and MIMIC.
2.	The authors should conduct experimental validation on a broader and more advanced set of language models, such as GPT-2 or LLaMA3, to support their hypothesis.

**Ethical Concerns:**

["NO or VERY MINOR ethics concerns only"]

**Final Justification:**

This paper offers a degree of novelty and contribution. However, because the zero-shot inversion demonstration is not compelling, the work has notable limitations and insufficient practical utility in real-world scenarios. I therefore maintain my original score unchanged.

**Limitations:**

Yes

**Quality:**

3

**Strengths And Weaknesses:**

Strengths:
1.	This paper introduces a novel scenario based on the assumption that the embedding spaces of text encoders tend to converge. Under this assumption, it enables the transformation of embeddings from one model into the feature space of another.
2.	The authors propose the vec2vec framework to realize this scenario and demonstrate its effectiveness to some extent.
Weaknesses:
1.	The Strong Platonic Representation Hypothesis proposed by the authors for text models lacks extensive empirical validation. The authors only selected six models for embedding translation, which provides insufficient evidence to support the general applicability of this hypothesis across modern text encoders of varying sizes, architectures, and training data. In particular, there is a lack of evaluation on the GPT series, which are more widely used in the language domain.
2.	The core application of this work should be zero-shot inversion—recovering the original text represented by the source model’s embeddings. However, the results in this area are not very strong.

---

> ### Author Rebuttal · Authors · 2025-07-31
>
> Thank you for the helpful review. We attempted to address each of your comments below, broken down by weakness and question.
>
> ---
> **(W2, Q1):** Thank you for the suggestion, we are evaluating inversions on TweetTopic (to complement the existing evaluations on MIMIC) for the next version of our paper. While our zero-shot inversion numbers are not as strong as methods with access to the ground-truth encoder or other auxiliary information (e.g., [1, 2]), we emphasize that our setting is different and harder.
>
> Prior to our work, **it was not known whether anything *at all* can be extracted from embedding vectors without prior information** (i.e., with no supervision, no paired data, and no knowledge of the encoder). That zero-shot inversion is even possible in this setting is a significant result.
>
> Even though we use zero-shot inverters designed for non-noisy embeddings, they still extract sensitive information from our noisy translations. Figures 5 and 6 show that inverted embeddings leak names, entities, and IP. Our experiments show that as many as 80% of inversions reveal confidential information.
>
> Furthermore, sensitive information can also be revealed using zero-shot attribute extraction directly on translated embedding vectors. As we show in Table 5, our translations reveal nearly as much sensitive information about medical records and corporate emails as the ground-truth embeddings. We expect that future work on better inversion methods in the unsupervised, unpaired translation setting will demonstrate even more and finer-grained information leakage from embeddings.
>
> We will reinforce the above points in the introduction and discussion sections.
>
> [1] Morris, et al. “Text Embeddings Reveal (Almost) As Much As Text”. 2023.
>
> [2] Zhang, et al. “Universal Zero-shot Embedding Inversion”. 2025.
>
> ---
> **(W1, Q2):** The six models in our evaluation cover a broad range of realistic embedding models:
> * Six different producers / organizations / distinct training datasets and regimes
> * Three standard encoder sizes: ~100M, ~150M, and ~300M
> * Four different architectures: T5, BERT, RoBERTa, CLIP
> * Unimodal and multimodal (CLIP) training
> * English-only and multilingual (Granite) training
> * Four “vintages”: 2021-2024 (eg, Granite is more recent than Llama3).
>
> Four of the models in our evaluation (E5, GTE, Stella, and Granite) actually **outperform** Llama3 on both Multilingual and English MTEB.  GTR performs comparably to Llama3. CLIP and GPT2 are not measured. Each of our models was near-SotA at the time of their release.
>
> Due to computational constraints (see Appendix A for details), we could not perform a comprehensive evaluation on 7B models. We agree that it is interesting to evaluate our methods on bigger models and are currently training vec2vec on 4B Qwen embeddings (which are newer and more commonly used than GPT2 and Llama3).

---

> ### Author Response · Authors · 2025-08-07
> **Results Update**
>
> As promised in our rebuttal above, we have been working on a few experiments and have a few preliminary updates to share.
>
> ---
>
> **Bigger Models:** We have been investigating whether or not our findings generalize to other, larger embeddings (in particular the state-of-the-art Qwen models), and in preliminary tests we were able to achieve the following ranks:
>
> - GTE-Qwen-2 2B → Qwen-3 4B: **1.041**
> - Qwen-3 4B → GTE-Qwen-2 2B: **1.037**
>
> Importantly, while the model producers are the same (Alibaba / Qwen), the architectures are very different. As the name suggests, GTE-Qwen2 Encoder’s backbone model is GTE, while Qwen3 Encoder’s backbone is a dense version of their MOE model Qwen3. Hence, these findings,
>
> 1. are cross-backbone,
> 2. and show that our findings generalize to (significantly) larger models.
>
> We are also currently training a (base) GTE→ Qwen-3 4B translator which is currently achieving a rank of **~80** (already significantly better than the baselines). This model *is still training*, and we expect these numbers to continue to improve. For the next version of our paper, we will have results for more pairs of models.
>
> ---
>
> **TweetTopic Zero-Shot Inversion:** In addition, we evaluated zero-shot inversion on TweetTopic. Ala Figure 5, we calculate the leakage of confidential information in a random 128-tweet sample via inversion as judged by an LLM. Cells denote judge accuracy.
>
> ```
> Percentage of Tweets With Leakage
>
> M1\M2           e5 granite     gte     gtr  stella
> e5             N/A    43.0    42.2    27.3    46.9
> granite       50.8     N/A    51.6    34.4    51.6
> gte           46.1    37.5     N/A    39.1    62.5
> gtr           39.1    48.4    50.0     N/A    53.9
> stella        57.0    42.2    67.2    46.9     N/A
> ```
>
> As shown, our zero-shot inversions leak confidential information in as high as 67.2% of tweets. We will expand on these results in the next version of our paper.

---

> > ### Comment · Reviewer_5c82 · 2025-08-09
> >
> > I sincerely appreciate your efforts on this paper; I have no additional questions.

---

### Decision · Program_Chairs · 2025-09-17

**Decision:**

Accept (poster)

**Comment:**

This paper considers a novel setting where we translate text embeddings between model spaces with no paired data or encoder access. The proposed vec2vec architecture (adapters + shared backbone with adversarial, cycle‑consistency, and vector‑space‑preservation losses) is a thoughtful adaptation of unpaired translation to the representation domain, with ablations indicating each term matters. Empirically, results are strong across multiple backbone families (BERT/T5/RoBERTa) and hold reasonably OOD. Main reservations are training instability and compute, some lingering over‑claiming tone vs. CycleGAN/word‑alignment precedents, and initially sparse details on OT comparisons (partly addressed during discussion).